# Discharge hydrograph estimation at upstream-ungauged sections by coupling a Bayesian methodology and a 2D GPU Shallow Water model

Alessia Ferrari[1], Marco D'Oria[1], Renato Vacondio[1], Alessandro Dal Palù[2], Paolo Mignosa[1], and Maria Giovanna Tanda[1]

[1]Department of Engineering and Architecture, University of Parma, Parma, Italy
[2]Department of Mathematical, Physical and Computer Sciences, University of Parma, Parma, Italy

*Correspondence to:* Alessia Ferrari (alessia.ferrari@unipr.it)

**Abstract.** This paper presents a novel methodology for estimating the unknown discharge hydrograph at the entrance of a river reach, when no information is available. The methodology couples an optimization procedure, based on the Bayesian Geostatistical Approach (BGA), with a forward self-developed 2D hydraulic model. In order to accurately describe the flow propagation in real rivers characterized by large floodable areas, the forward model solves the 2D Shallow Water Equations (SWEs) by means of a Finite Volume explicit shock-capturing algorithm. The 2D-SWE code exploits the computational power of Graphics Processing Units (GPUs), achieving ratio of physical to computational time up to 1000. With the aim of enhancing the computational efficiency of the inverse estimation, the Bayesian technique is parallelized developing a procedure based on the Secure Shell (SSH) protocol that allows to take advantage of remote High Performance Computing clusters (including those available on the Cloud) equipped with GPUs. The capability of the methodology is assessed by estimating irregular and synthetic inflow hydrographs in real river reaches, taking into account also the presence of downstream corrupted observations. Finally, the procedure is applied to reconstruct a real flood wave in a river reach located in Northern Italy.

## 1 Introduction

The definition of discharge hydrographs in specific river sections is still a relevant hydraulic problem not only for flood modelling purposes, but also for more practical issues related to flood protection measures, hydropower plants, water resource management, design of new structures, etc. Flood routing techniques, either hydrological or hydraulic, are extensively studied and widely used to estimate discharge hydrographs in downstream ungauged sites based on data available at upstream gauged stations (forward propagation). However, often, the flow hydrograph is required in a river section that is completely ungauged and does not have upstream useful information for its definition. In these cases, discharge hydrographs at specific sites can be estimated by coupling rainfall-runoff and forward flood propagation models. However, rainfall-runoff models (Beven (2011)) present several uncertainties associated, for example, with the choice of the model for the basin schematization, with the evaluation of the effective rainfall, and with the calibration procedure. An alternative approach is to assess the upstream unknown flow hydrograph using only the information, in terms of discharge values or water levels, available downstream the

selected site and, possibly, the characteristics of the river reach. In the literature, this approach is known as reverse flow routing (D'Oria and Tanda (2012)), an ill-posed inverse problem that presents two main challenges: the solution may be non-unique, and instabilities may arise during the inversion. The traditional attempts of solving the reverse flow routing problem are based on two main approaches: the solution of a reverse form of the Saint Venant equations (e.g. Eli et al. (1974), Szymkiewicz

(1993), Dooge and Bruen (2005), Bruen and Dooge (2007)) and the back oriented application of hydrological routing schemes (e.g. Das (2009), Koussis et al. (2012), Koussis and Mazi (2016)). Beyond the approximations introduced by the hydrological routing schemes, the above procedures were applied to simplified reach geometries and flow conditions. In almost all cases, especially considering downstream information affected by errors, instabilities and spurious oscillations appeared; low-pass filters, with subjective parameters, were sometime used to damp the estimated inflow fluctuations. D'Oria and Tanda (2012)

and Zucco et al. (2015) provide additional references and details on the reverse flow routing problem.

In addition to the above procedures, the estimation of an unknown upstream flow hydrograph, based only on downstream information (observations), can be performed via optimization methods. These techniques aim at finding the upstream flow hydrograph that, routed downstream, best matches the available observations. D'Oria and Tanda (2012) solved the reverse flow routing problem adopting, as optimization procedure, a novel Bayesian Geostatistical Approach (BGA) that considers the

15 flow hydrograph as a continuous random function that presents autocorrelation. The authors showed the capability of the BGA methodology, in combination with a forward hydraulic model, to estimate the discharges in an upstream ungauged section based only on an available downstream flow hydrograph: the solution was stable also in the presence of corrupted downstream flow values. The forward model, which solves the 1D Saint Venant equations, was considered already implemented and calibrated and able to describe, with sufficient accuracy, the hydraulic routing process. The BGA method was further extended in

order to adopt, as downstream observations, stage hydrographs instead of discharge ones (D'Oria et al. (2014)). Saghafian et al. (2015) identified the upstream hydrograph of a river reach, given the downstream one, by using a Genetic Algorithm coupled with a forward hydraulic model that solves the 1D Saint-Venant equations under the kinematic wave approximations. Only some minor oscillations and instabilities occurred during the inversion, but the Authors applied the procedure to a rectangular prismatic channel and no errors were added to the downstream observations. Zucco et al. (2015) investigated the reverse flow

routing process in natural channels, and estimated the discharge hydrograph in ungauged sections, by means of a Genetic Algorithm coupled with a simplified routing model. The parametric forward model was based on the continuity equation written in a characteristic form, lumped over the entire river reach, and on simplified rating curves at the channel ends. In addition, the unknown inflow hydrograph was assumed distributed in time as a Pearson type III function with three parameters, thus preventing the possibility of estimating real flood waves with irregular shapes (e.g. multi-peak hydrographs).

All the previously cited works adopted 1D hydraulic models or simplified hydrological routing schemes, in combination with different optimization procedures. Nevertheless, in many real cases, the complex hydrodynamic field generated by the flood propagation cannot be accurately described under 1D assumptions and it is necessary to adopt schemes based on the 2D Shallow Water Equations, even if this poses the drawback of the computational burden and requires a detailed terrain survey. However, nowadays, bathymetric data can be easily obtained from high-resolution Digital Terrain Models (DTM) and fast 2D

numerical models have been developed. With the purpose of estimating the discharge hydrograph in an upstream ungauged

river section, having water level information only in a downstream observation site, this paper extends the BGA methodology for reverse flow routing of D'Oria and Tanda (2012) and D'Oria et al. (2014) to a 2D forward algorithm in order to model natural rivers with complex geometry, including flood plains and floodable areas. With this aim, the stable, accurate and fast PARFLOOD GPU code (Vacondio et al. (2014),Vacondio et al. (2017),Vacondio et al. (2016)), which solves the conservative form of the 2D Shallow Water Equations on a finite volume scheme, is adopted as forward model and coupled to the inverse estimation procedure. In order to reduce the computational time, the Jacobian matrix estimation procedure, which is the key point of the BGA method, has been parallelized. Additionally, a host-server data management procedure has been implemented, so as to exploit the computational power of remote large modern supercomputer and/or cloud HPC resources. The capability of the optimization procedure has been tested by estimating real or pseudo-real inflow hydrographs in natural river reaches, where 1D models cannot accurately describe the flood propagation. Moreover, during the discharge estimation, the presence of downstream corrupted observations has also been taken into account, since registered data at gauging stations are quite often affected by instrumental errors.

The paper is organized as follows: in Sect. 2 the theory of the Bayesian Geostatistical Approach is illustrated. A step-by-step description of the inverse procedure is provided in Sect. 3: the parallel implemented scheme, the forward model optimization for reducing the run times and the iteration management between the local host and the remote server are described in detail. Section 4 is dedicated to the application of the procedure to synthetic test cases concerning the estimation of inflow hydrographs with different shapes in two rivers in Northern Italy. The practicability of the inverse procedure for reconstructing a historical flooding event is presented in Sect. 5. Some concluding remarks are finally outlined in Sect. 6.

## 2    Theory of the Bayesian Geostatistical Approach

The optimization software adopted to solve the reverse flow routing problem is the bgaPEST (Fienen et al. (2013)), which implements the Bayesian Geostatistical Approach of Kitanidis (1995) and it is developed according to the PEST (Model Independent Parameter Estimation) parameter estimation software (Doherty (2016)). The bgaPEST is appropriate for solving inverse problems (in a context of a highly parametrized inversion), which are characterized by unknown parameters that are correlated one another in space or time, as for example the discharge values of a flow hydrograph. The first applications of the inverse methodology were related to the estimation of spatial parameter fields in a groundwater context (Kitanidis and Vomvoris (1983), Hoeksema and Kitanidis (1984), among others) but later the method was adopted to evaluate unknown time functions in different areas (e.g. Snodgrass and Kitanidis (1997), Michalak et al. (2004), Butera et al. (2013), D'Oria and Tanda (2012), D'Oria et al. (2015), Leonhardt et al. (2014)).

### 2.1    Bayes' theorem

The crux of the adopted bgaPEST, as well as other methods based on the Bayesian Approach, is Bayes' theorem, which reads:

$$p\left(\mathbf{s}|\mathbf{y}\right) \propto L\left(\mathbf{y}|\mathbf{s}\right)p\left(\mathbf{s}\right), \tag{1}$$

where $\mathbf{s}$ is the vector of the unknown parameters, $\mathbf{y}$ is the vector of the measured data, $p(\mathbf{s}|\mathbf{y})$ is the posterior probability density function (pdf) of $\mathbf{s}$ given $\mathbf{y}$, $L(\mathbf{y}|\mathbf{s})$ is the likelihood function and p($\mathbf{s}$) is the prior probability density function of $\mathbf{s}$. Since the present work aims at estimating an upstream hydrograph in an ungauged section, assuming the knowledge of downstream water levels, $\mathbf{s}$ represents the discharge values over time of the unknown inflow hydrograph, whereas $\mathbf{y}$ denotes the downstream water level observations. Following Eq.(1), the posterior pdf can be seen as a combination between a priori knowledge on the parameters (prior pdf), where a priori means that the observed data are still not considered, and information about parameters contained in the measured data (likelihood function) (Glickman and Van Dyk (2007)). In the BGA method proposed by Kitanidis (1995), the prior pdf and the likelihood function are described by means of Gaussian distributions and the best set of parameter $\mathbf{s}$ is obtained by maximizing the posterior pdf.

### 2.1.1 The likelihood function

The likelihood function $L(\mathbf{y}|\mathbf{s})$ in Eq. (1) characterizes the deviation between observed data and model results (Fienen et al. (2013)). Starting from the results of the forward model, $L(\mathbf{y}|\mathbf{s})$ delineates how a particular set of parameters $\mathbf{s}$ is able to reproduce the observations $\mathbf{y}$ in space and/or time, thus accounting for the epistemic errors. The investigated inverse problem presents different sources of errors that are related to the conceptual schematization of the inverse procedure, to the numerical forward model and to the data measurement. In the likelihood function, the errors are assumed to be independent and identically distributed, with zero mean and covariance matrix expressed as follows:

$$\mathbf{R} = \sigma_R^2 \mathbf{I}, \tag{2}$$

where $\sigma_R^2$ denotes the variance that expresses the misfit between observed and modeled data, and $\mathbf{I}$ is the identity matrix.

### 2.1.2 The prior probability density function

The prior knowledge about $\mathbf{s}$ (p($\mathbf{s}$) in Eq.(1)) is limited to the definition of a mean value (unknown and estimated during the procedure) and a characteristic about the continuity and/or smoothness of the parameter field described through a covariance function. It furnishes a soft knowledge about the structure/shape of the unknowns and provides a regularization of the solution; the prior pdf can also be used to enforce non-negativity to the parameters (D'Oria and Tanda (2012)). The prior mean is defined as:

$$E[\mathbf{s}] = \mathbf{X}\boldsymbol{\beta}, \tag{3}$$

where $E$ is the expected value, $\boldsymbol{\beta}$ is the vector of drift coefficients, and $\mathbf{X}$ is a known matrix of basis functions. In our case $\boldsymbol{\beta}$ is a single unknown scalar, but different drift coefficients can be used to introduce discontinuities in the stochastic function to be estimated (e.g. when the unknown parameters are likely to form distinct populations). For example, in the context of reverse flow routing problems, multiple values of $\boldsymbol{\beta}$ are adopted if more than one inflow hydrograph must be estimated at the same time (e.g. the inflow on both the upstream branches of a river confluence). The matrix of basis function, $\mathbf{X}$, links each unknown

parameter with the corresponding element of $\boldsymbol{\beta}$ and, at the same time, specifies the model of the mean (e.g. constant mean, mean with a trend, etc.); in our case the mean is constant and therefore $\mathbf{X}$ is a single vector of ones (Fienen et al. (2008)).

The prior covariance matrix of the unknown parameters $\mathbf{Q}_{ss}$ is then defined as:

$$\mathbf{Q}_{ss} = E\left[(\mathbf{s} - \mathbf{X}\boldsymbol{\beta})(\mathbf{s} - \mathbf{X}\boldsymbol{\beta})^T\right]. \tag{4}$$

In the context of geostatistics, the covariance matrix $\mathbf{Q}_{ss}$ is a function of the separation distance (in time in this case) between the parameters and describes their deviations from the mean behavior. Different functions can be adopted to describe the covariance; for example, it can be assumed as a linear function, represented through a limiting case of the exponential covariance function (Fienen et al. (2008)), according to the following relation:

$$\mathbf{Q}_{ss}(\theta) = \theta l \exp\left(-\frac{|\mathbf{d}|}{l}\right), \tag{5}$$

where $\mathbf{d}$ represents the vector of the separation times between all the parameter pairs ($d_{i,j} = t_i - t_j$ with $i,j = 1, \ldots, N_p$, $t$ denoting the time associated with each parameter and $N_p$ the total number of unknowns), $l$ a fixed integral scale ($l = 10\max(d)$) and $\theta$ the slope (structural parameter) that governs the correlation between the discharge values of the unknown hydrograph. A different formulation (D'Oria et al. (2014)) defines the prior covariance matrix $\mathbf{Q}_{ss}$ by means of a Gaussian function characterized by two structural parameters ($\sigma_s^2$ and $l$):

$$\mathbf{Q}_{ss}(\sigma_s^2, l) = \sigma_s^2 \exp\left(-\frac{|\mathbf{d}^2|}{l^2}\right), \tag{6}$$

where $\sigma_s^2$ denotes the variance. The linear function (Eq.(5)) enforces only continuity to the solution whereas the Gaussian function (Eq.(6)) adds also some degree of smoothness, but the final solution is still driven by the observations.

### 2.1.3  The posterior probability density function

With the assumptions made, the likelihood and prior terms that compose the posterior pdf of Eq.(1) can be rewritten as follows
(Fienen et al. (2009); D'Oria and Tanda (2012); D'Oria et al. (2014)):

$$L(\mathbf{y}|\mathbf{s}) = \exp\left(-\frac{1}{2}(\mathbf{y} - \mathbf{h}(\mathbf{s}))^T \mathbf{R}^{-1}(\mathbf{y} - \mathbf{h}(\mathbf{s}))\right) \tag{7}$$

$$p(\mathbf{s}) = \exp\left(-\frac{1}{2}(\mathbf{s} - \mathbf{X}\boldsymbol{\beta})^T \mathbf{Q}_{ss}^{-1}(\mathbf{s} - \mathbf{X}\boldsymbol{\beta})\right). \tag{8}$$

The term $\mathbf{h}(\mathbf{s})$, in the likelihood function, represents the modeled values in the same place and time as the available observa-
tions $\mathbf{y}$. Therefore, to evaluate $\mathbf{h}(\mathbf{s})$, a forward model of the considered river reach that is able to describe the hydraulic routing process is required in order to provide, for a given set of parameter $\mathbf{s}$, the corresponding downstream water levels.

Recalling that the aim of the inverse procedure is to obtain the vector of the unknown parameters $\mathbf{s}$, as well as to quantify the uncertainty in the estimation, the solution is found by maximizing the posterior pdf or, more conveniently, minimizing its negative logarithm (objective function) (Fienen et al. (2013)).

In case a linear relationship between parameters and observations (linear forward model) holds, a computationally efficient method to find the best estimate $\hat{\mathbf{s}}$ of vector $\mathbf{s}$ (and $\hat{\boldsymbol{\beta}}$ of $\boldsymbol{\beta}$) is obtained introducing the vector $\boldsymbol{\xi} = (\mathbf{HQ}_{ss}\mathbf{H}^T + \mathbf{R})^{-1}(\mathbf{y} - \mathbf{HX}\hat{\boldsymbol{\beta}})$ and solving the following linear system of equations (Fienen et al. (2009)):

$$\begin{cases} \hat{\mathbf{s}} = \mathbf{X}\hat{\boldsymbol{\beta}} + \mathbf{Q}_{ss}\mathbf{H}^T\boldsymbol{\xi} \\ \begin{bmatrix} \mathbf{HQ}_{ss}\mathbf{H}^T + \mathbf{R} & \mathbf{HX} \\ \mathbf{X}^T\mathbf{H}^T & 0 \end{bmatrix} \begin{bmatrix} \boldsymbol{\xi} \\ \hat{\boldsymbol{\beta}} \end{bmatrix} = \begin{bmatrix} \mathbf{y} \\ 0 \end{bmatrix}, \end{cases} \tag{9}$$

where $\mathbf{H}$ is the sensitivity (Jacobian) matrix, representing how the observations $\mathbf{y}$ are influenced by each unknown parameter $s_i$ (D'Oria et al. (2015)). However, for the particular problem under investigation, $\mathbf{h(s)}$ is non-linear and therefore matrix $\mathbf{H}$ depends on $\mathbf{s}$. Following the quasi-linear geostatistical approach (Kitanidis (1995)), the relationship between observations and parameters can be successively linearized about a candidate solution $\mathbf{s}_k$, where $k$ represents each iteration:

$$\mathbf{h(s)} \approx \mathbf{h(s}_k) + \widetilde{\mathbf{H}}_k\left(\mathbf{s} - \mathbf{s}_k\right), \tag{10}$$

and then a correction to the measurements is applied according to the following relation:

$$\mathbf{y}_k = \mathbf{y} - \mathbf{h(s}_k) + \widetilde{\mathbf{H}}_k\mathbf{s}_k. \tag{11}$$

Therefore, the sensitivity matrix is evaluated at each iteration as follows (D'Oria et al. (2014)):

$$\widetilde{\mathbf{H}}_k = \left.\frac{\partial \mathbf{h(s)}}{\partial \mathbf{s}}\right|_{\mathbf{s}_k}. \tag{12}$$

Analogously to the linear system in Eq. (9), the linearized system is solved according to:

$$\begin{bmatrix} \widetilde{\mathbf{H}}_k\mathbf{Q}_{ss}\widetilde{\mathbf{H}}_k^T + \mathbf{R} & \widetilde{\mathbf{H}}_k\mathbf{X} \\ \mathbf{X}^T\widetilde{\mathbf{H}}_k^T & 0 \end{bmatrix} \begin{bmatrix} \boldsymbol{\xi}_{k+1} \\ \hat{\boldsymbol{\beta}}_{k+1} \end{bmatrix} = \begin{bmatrix} \mathbf{y}_k \\ 0 \end{bmatrix}, \tag{13}$$

and the next estimate of the parameters is evaluated by means of:

$$\widetilde{\mathbf{s}}_{k+1} = \mathbf{X}\hat{\boldsymbol{\beta}}_{k+1} + \mathbf{Q}_{ss}\widetilde{\mathbf{H}}_k^T\boldsymbol{\xi}_{k+1}. \tag{14}$$

A proper selection of the covariance function structural parameters ($\theta$, $\sigma_s^2$ and $l$) and optionally of the epistemic error variance $\sigma_R^2$ is important in order to reach a good solution. The structural parameters are estimated from the data using a Bayesian adaptation of the Restricted Maximum Likelihood (RML) method of Kitanidis (1995) that allows reaching the best compromise between the fitting of the modeled data with the observations and the prior information (Fienen et al. (2013)). Dealing with non-linear problems, unknowns ($\mathbf{s}$) and structural parameters must be iteratively estimated in successive steps. The linearization process ends if the improvement (absolute difference between two successive iterations) in the objective function is below a user defined value or if the maximum number of iterations $N_i$ is reached. The structural parameter iteration loop (outer loop) progresses until the $L_2$-norm of the differences between structural parameter values at consecutive iterations is below a user defined value or if the maximum number of iterations $N_o$ is reached (Fienen et al. (2013)). Finally, at the end of the estimation, the linearized uncertainties of the unknowns can be evaluated in terms of the posterior covariance matrix of the estimated parameters (Fienen et al. (2013)). The diagonal elements of this matrix represent the posterior variance ($\sigma^2$) of the estimated parameters and thus, the 95% credibility interval of the solution is evaluated as $\pm 2\sigma^2$.

## 3    Description of the Bayesian estimation procedure

After having described the theory of the Bayesian Geostatistical Approach in Sect. 2, some operational information about the BGA inverse procedure is now illustrated. As mentioned in the Introduction and sketched in Fig. 1-a, the goal of the adopted BGA methodology is the estimation of the discharge hydrograph in an upstream-ungauged river section (identified
by a question mark in Fig. 1-a), having information about water levels observed in a downstream section (intermediate site in Fig. 1-a). A boundary condition, downstream of the observation site, must also be specified; this can be based on observed data or can be approximated extending the computational domain faraway from the intermediate section. The inverse method estimates $N_p$ parameters (the vector of the unknown parameters **s** in Eq. (1)) that originates from the discretization of the upstream discharge hydrograph by means of time intervals, regular in this case (Fig.1-b).

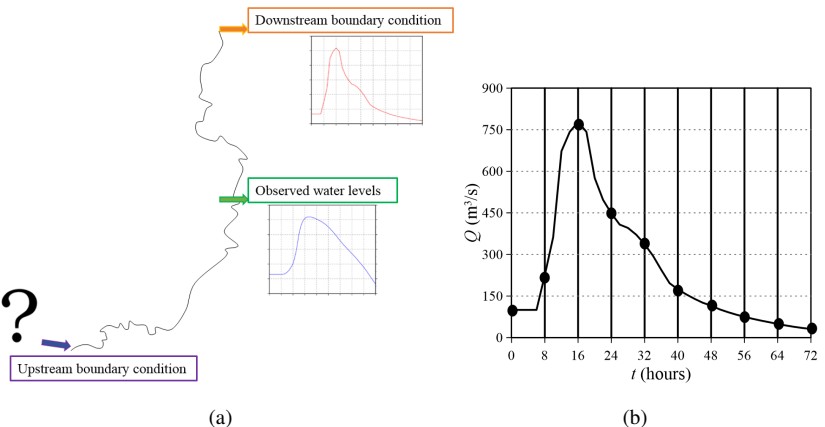

(a)                                                              (b)

**Figure 1.** Definition of the reverse flood routing problem (a) and of the unknown parameters (b).

The BGA algorithm solves the inverse problem by means of the following steps.

First, the unknown parameters and the structural ones are initialized. The first ones may be all assumed equal to a constant discharge value coherent with the considered river, whereas the starting values for the structural parameters are assigned so that the variability between contiguous parameters is small (flat solution, with a high degree of correlation); complexity is then introduced during the optimization process if supported by the data. The variance of the epistemic errors is assumed close to
the expected one.

Assuming the first guess of the unknown parameters as upstream boundary condition, the hydraulic forward model is run and the resulting water levels are extracted at the observation site. The simulation of a base run, once a particular set of parameters has been assumed (deriving from the initialization or from previous estimation steps), represents a mandatory step for the Jacobian matrix evaluation, which is performed at this point of the procedure in order to quantify how each observation is
influenced by the variation of each estimable parameter. Particularly, Eq. (12) is approximated using a finite difference method, and hence each element of the matrix is evaluated as the ratio between the variation of each observation (numerator) for given variation of each parameter (denominator) with respect to the base run. Therefore, additionally to the base run, the hydraulic

forward model is further run as many times as the number of parameters to estimate $N_p$. At each run, a single value of the upstream boundary condition is modified by a known quantity with respect to the previous value, and the hydraulic forward model is run again. As a consequence, each simulation tests the sensitivity of the resulted water levels (all the observations at once) to the variation of a single parameter $i$.

5 In order to exemplify this step, Fig. 2-a shows the discharge imposed as upstream boundary condition for a base run of an intermediate set of parameters: after the propagation, the resulting water levels extracted at the observation site are shown in Fig. 2-c. To test the sensitivity to parameter $i$, in Fig. 2-b the considered parameter is changed by a known quantity and a new upstream boundary condition is defined (solid line); it is worth noting that the solid and the dotted lines differ only for the parameter $i$. The water levels resulting from this single parameter variation are shown in Fig. 2-d (solid line): they are

10 identical to the base run ones until time $i$-1, whereas after that time they differ from those of the base run (dotted line). The computation of the differences between the resulting water levels of the simulation $i$ and of the base run (solid and dotted lines) and the variation of parameter $i$ allows computing the column $i$ of the Jacobian matrix, which is a $N_{obs} \times N_p$ matrix, where $N_{obs}$ represents the number of the observations. After $N_p$ runs, the Jacobian/sensitivity matrix is evaluated and a new set of parameters **s** is estimated (Eq. (14)).

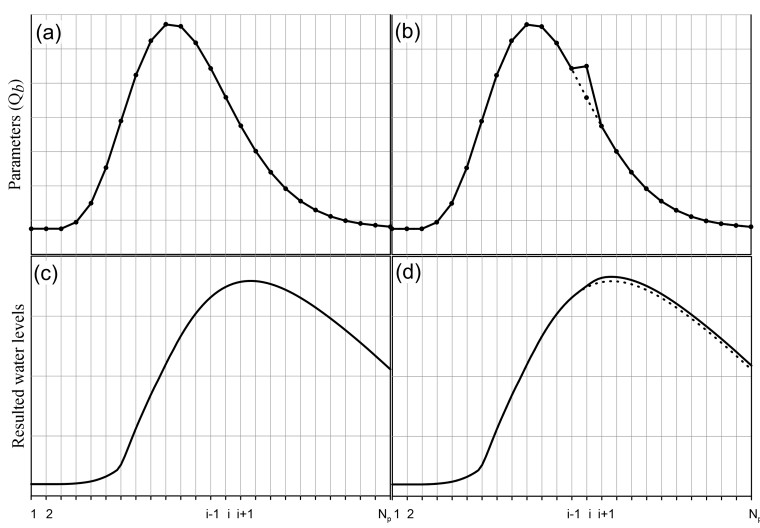

**Figure 2.** Example of the base run (a) and of the run $i$ for the Jacobian matrix evaluation (b).

15 This procedure is repeated until convergence or the maximum number of iteration $N_i$ is reached. Then, the structural parameters are estimated using the last set of parameters **s**. Due to the non-linearity of the forward problem, the model and the structural parameter estimation is repeated until convergence or the maximum number of iterations $N_o$ is reached. Therefore, the BGA implementation requires running the forward model $N_t$ times, according to the following relation (Fienen et al. (2013)):

20 $N_t = (N_p + 1) N_o N_i + 1.$            (15)

The whole BGA procedure previously described is sketched in Fig. 3-a.

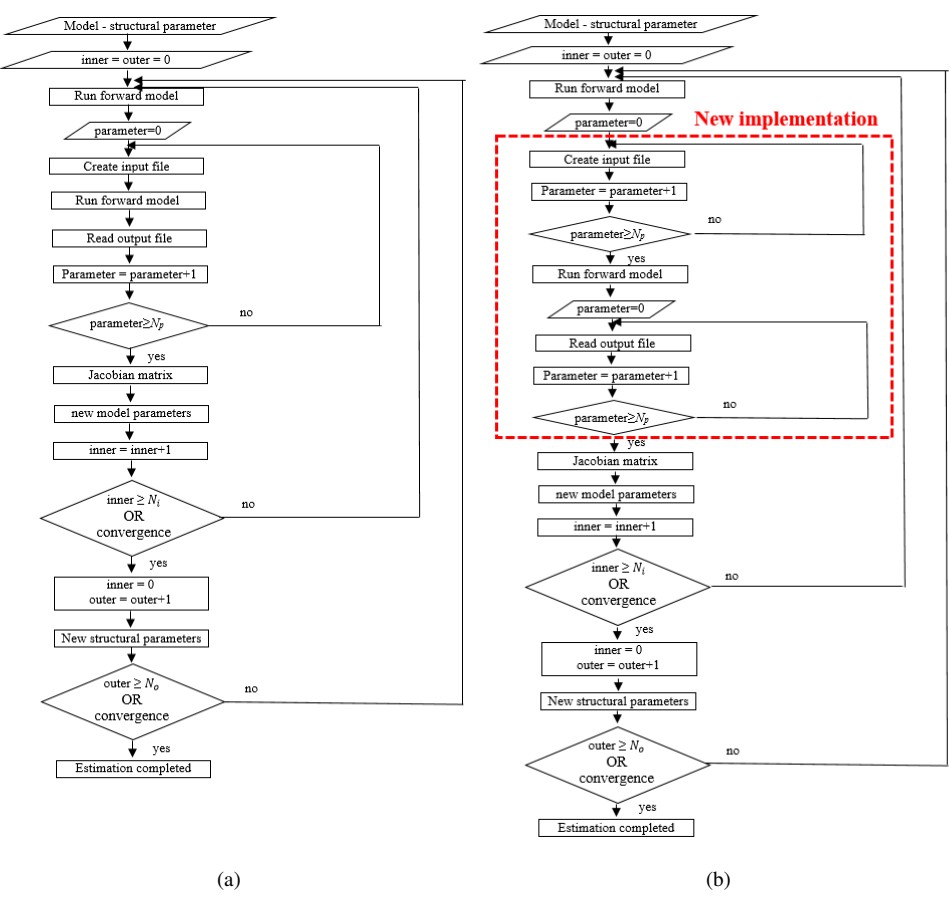

**Figure 3.** Scheme of BGA algorithm in the serial (a) and parallel (b) version.

## 3.1  Parallelization of the Jacobian matrix evaluation

The most relevant contribution to the total computational time required by the inverse procedure is ascribed to the forward model runs (i.e. the computation of each element of the Jacobian matrix), rather than to the bgaPEST operations. However, since each of the $N_p$ runs in Eq. (15) checks the sensitivity of the observations to the variation of a single parameter, the solution of a run does not affect the other ones. Therefore, in order to reduce the computational burden, the independent $N_p$ runs can be potentially performed in parallel.

In this work, the PARFLOOD 2D-GPU numerical model presented in Vacondio et al. (2014) and Vacondio et al. (2017) has been adopted for routing the inflow hydrograph. Therefore, the bgaPEST routine to evaluate the Jacobian matrix has been parallelized in order to take advantage of the computational capability of modern High Performance Computing (HPC) clusters, which are usually equipped with many GPUs. The implemented parallel procedure, which is sketched in the flow chart of Fig.

3-b, handles the parallelism among host and GPUs by means of the Secure Shell network protocol (SSH) and manages the most operative parts of the parallelism (login, run, etc.) outside the bgaPEST code. In the serial version (Fig. 3-a), the crucial part of the Jacobian matrix evaluation consists in a do-loop over the parameters. Considering the parameter $i$, first the input file that will be read by the forward model is written, then the model is run and finally the resulting values are read. In the modified version (Fig. 3-b), this main loop is split in three parts: first, all the input files (equal to $N_p$), in each of which a particular parameter is modified, are written, then the forward model is run ($N_p$ times), and finally a second loop is performed to read all the resulted values.

## 3.2 The forward model

In the parallel bgaPEST (Fig. 3-b), the "Run forward model" instruction actually runs a shell script that controls the file transfer between the host (a standard PC or a single node of a cluster) and the HPC platform, the creation of the $N_p$ simulations for the Jacobian matrix evaluation, and the run of the 2D-SWE GPU code on the device (GPU). In the present work, a cluster with 10 NVIDIA ® Tesla ® P100 GPUs hosted by the University of Parma was adopted. As shown in Fig. 4, the bgaPEST algorithm runs on the CPU of a computer, where the $N_p$ simulations (in Fig. 4 assumed equal to three for the sake of simplicity) are first created and then sent to the server user partition, by means of the SSH protocol. Here, the cluster access node schedules all the jobs submitted by the users, using the HPC scheduler Portable Batch System (PBS). Then, each simulation is assigned to a specific GPU node. At the end of the computation, the observations are extracted and the output files remain on the cluster partition, until the CPU verifies via SSH the end of the simulation and copies the results back. The procedure sketched in Fig. 4 and following described represents the parallelization of the Jacobian matrix computation.

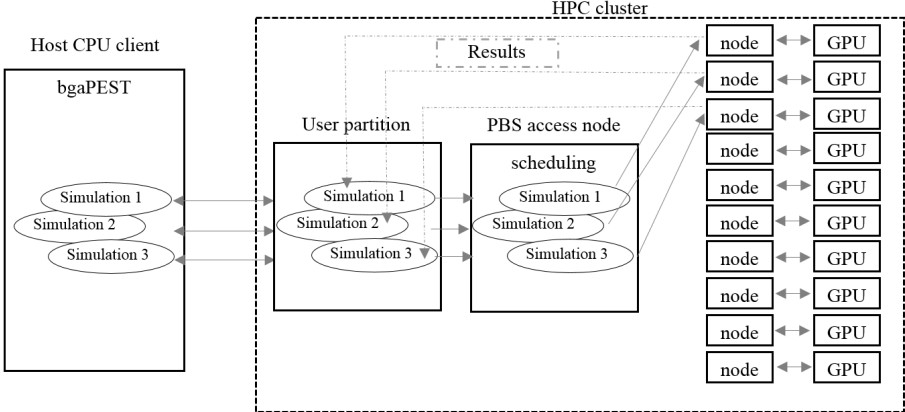

**Figure 4.** Schematization of the data transfer assuming three parameters and thus three parallel simulations.

Listing 1 provides a detailed description of the "Run forward model" shell file. In order to use the Algorithm for different test cases and potentially on different HPC clusters, all the paths are first declared together with the involved variables (number

of parameters to estimate, time interval among parameters, start/end of the simulation) (line 2). Then, the algorithm (line 3) checks if the considered run is one useful for the Jacobian matrix evaluation, where a given parameter varies, or if it is the base run. Considering the first *if* condition as true (line 3), the script generates and copies the input files for all the $N_p$ simulations to the server (lines 5-7). These files contain the same bathymetrical, initial conditions (water level and velocity) and roughness configuration, but a different upstream boundary condition; each simulation tests the sensitivity of the observations to the variation of a given model parameter. Moreover, all the simulations adopt the same grid (Cartesian or multiresolution), which is generated only once at the beginning of the procedure. It is relevant to note that all the $N_p$ simulations have not to be run from time $t_{start}$ to time $t_{end}$; in fact, the variation of parameter $i$ affects the observations only after time $t_{i-1}$ (see Fig. 2). The PARFLOOD model guarantees the possibility of using the results of the base run and starting simulations from time $t_{i-1}$. The theoretical physical time $T$ required to evaluate the Jacobian matrix simulating each of the $N_p$ runs from $t_{start}$ to $t_{end}$ is equal to:

$$T = N_p (N_p - 1) \Delta t, \tag{16}$$

where $\Delta t$ denotes the constant time interval between two consecutive parameters.

Conversely, the physical time $T^*$ required to simulate all the $N_p$ runs restarting the *i*-th simulation from time $t_{i-1}$ instead of $t_{start}$ is equal to:

$$T^* = (N_p - 1)\Delta t + \sum_{i=2}^{N_p} [N_p - (i-1)] \Delta t. \tag{17}$$

As pointed out by Eq. (16)-(17) and exemplified in Fig. 5, this simple operation allows reaching a relevant decrease of the total computational time. Therefore, at line 8, the algorithm computes the time useful to restart the simulation.

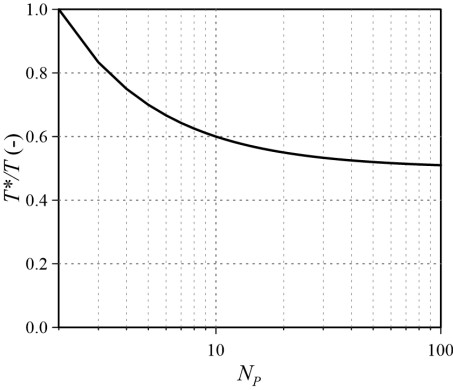

**Figure 5.** Time reduction $T^*/T$ as a function of the number of estimable parameters (the *x*-axis is in logarithmic scale).

In order to perform the simulation, the host logs in to the HPC cluster by means of the SSH protocol (line 9) and a sleep condition ensures the login procedure (line 10). Then the job is submitted to the queue of the cluster using external parameters for passing the name of the simulation folder and the time for restart (line 11): the submitted job contains the reference to the

PBS queue and the link to the executable 2D-SWE GPU code. At the end of the simulation, the water levels at the observation site are automatically extracted. Once the job is submitted, the SSH login is closed (line 12). After having submitted all the simulations, for each parameter (line 15) the code regularly (line 18) tests via SSH the presence of the end_file, which states the end of the simulation (line 20), and waits in case it is missing (line 25). Once the simulation is finished, the resulting observations are copied back to the Host client (line 28) and the folder is removed from the server (line 29).

Conversely, the *else* condition (line 30) is true for the base run. The simulation folder with the input files is copied to the server (line 31) and the job is submitted (line 34). Then, the algorithm periodically verifies the end of the simulation and copies the results back to the Host client (lines 39-49). It is relevant to note that the base run is performed first, whereas the other $N_p$ ones can be performed in parallel.

**Listing 1.** "Run forward model" for the parallel bgaPEST scheme

```bash
#!/bin/bash
Variable and path declaration
if [parameter run];
then
for (( i=1; i<=Np; i++ ))
do
Create and copy the simulation folder to the server (simulation_i)
Compute the time from which restarting the simulation
ssh server name << EOF
sleep 15
ssh submission: frame number for restart , name of the test case, job to submit
exit
EOF
done
for (( i=1; i<=Np; i++ ))
do
end_file=0
while [ end_file -eq 0 ];
do
ssh server name   find server_path -iname end_file.txt | wc -l > end_file
if [end_file];
then
continue
else
sleep 10
fi
done
Copy the file with observation from server to CPU
Remove the simulation folder on the cluster
else
Create and copy the simulation folder to server
```

```
ssh server name << EOF
sleep 15
ssh submission: frame number for restart , name of the test case , job to submit
exit
EOF
end_file=0
while [ end_file −eq 0 ];
do
ssh server name   find server_path −iname end_file.txt | wc −l > end_file
if [end_file];
then
continue
else
sleep 10
fi
done
Copy the file with observation from server to CPU
fi
```

## 4    Application of the inverse methodology to synthetic test cases

In the context of applying the BGA method described above, it is worth noting that reference solutions for inverse problems are by definition unavailable, since the goal of the methodology is the estimation of an upstream inflow hydrograph that is unknown at the beginning of the process. Therefore, in this section the inflow hydrographs in two natural rivers in Northern Italy are estimated and the reference solutions, which are necessary in order to validate the inverse procedure, are obtained as follows (D'Oria et al. (2014)). Considering the domain in Fig. 6, a selected inflow discharge $Q^{ref}$ is routed from the upstream section A to the downstream boundary D, where a rating curve is imposed far away from C. The resulting water level hydrographs are extracted at sites B and C. The inverse procedure is then applied to the sub-domain sketched with solid line in Fig. 6, by assuming the water levels in sites B and C (derived in step 1) as observations and downstream boundary condition, respectively. The information in sub-reach C-D is only preparatory for setting up the synthetic cases and it is not used in the inverse procedure. Imposing a rating curve in D allows to obtain water levels with a non-unique stage-discharge relationship in section C, which is more close to the real situations when applying the inverse procedure. The methodology estimates the inflow $Q^{est}$ assuming that no information is available on the discharge (or water stage) at the inflow section A.

Quantitative information about the accuracy of the inverse methodology is provided evaluating the differences between the reference $Q^{ref}$ and the estimated $Q^{est}$ hydrographs by means of three different indicators. First, the Nash-Sutcliffe efficiency criterion (Nash and Sutcliffe (1970)) $E_h$ was adopted, according to the following relation:

$$E_h = \left[1 - \frac{\sum_{i=1}^{N_p}(Q_i^{ref} - Q_i^{est})^2}{\sum_{i=1}^{N_p}(Q_i^{ref} - \overline{Q}^{ref})^2}\right] \cdot 100, \tag{18}$$

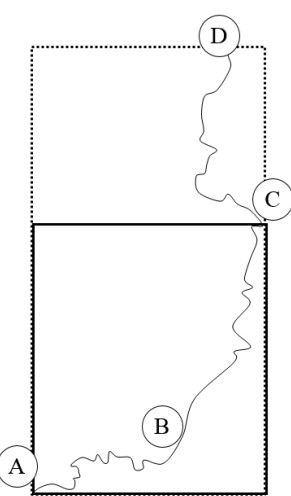

**Figure 6.** Exemplification of a test case definition.

where $N_p$ is the number of parameters, $Q_i^{ref}$ and $Q_i^{est}$ are the $i$-th reference and estimated inflow values, respectively, and $\overline{Q}^{ref}$ is the mean value of the reference hydrograph. Then, the root mean square error, RMSE, was evaluated as follows:

$$RMSE = \sqrt{\frac{\sum_{i=1}^{N}(Q_i^{ref} - Q_i^{est})^2}{N_p}}. \tag{19}$$

Finally, the estimation error in the peak discharge $E_p$ was assessed as:

$$E_p = \left[ \frac{Q_p^{est}}{Q_p^{ref}} - 1 \right] \cdot 100, \tag{20}$$

where $Q_p^{est}$ and $Q_p^{ref}$ denote the peak discharge value of the estimated and reference hydrographs, respectively.

### 4.1 Inflow hydrograph estimation on the Parma River

The first test concerns the estimation of a hypothetical discharge hydrograph at the entrance of the Parma River (Northern Italy). Figure 7-a illustrates the studied domain and the locations of the upstream boundary condition A, of the observation site B and of the downstream boundary section C. The domain includes a 20-km long embanked reach that is characterized by several meanders and flood plains. As shown in Fig. 7, the flow field significantly varies at low and high discharge values due to the river morphology. At the beginning of the flood wave, the flow is characterized by both low water depths (Fig. 7-b) and velocity (Fig. 7-d). Conversely, at the arrival of the flood peak, most of the meanders are cut by the flow, as shown in Fig. 7-c and Fig. 7-e for water depths and velocity, respectively. This makes the adoption of 1D numerical schemes not suitable to accurately describe the flood propagation.

The bathymetry was derived from a 1-m resolution DTM obtained through a LiDAR survey carried out in drought condition. The domain was discretized by means of a Cartesian grid with cell sizes $\Delta x = \Delta y = 4$ m and about $275 \cdot 10^3$ computing cells

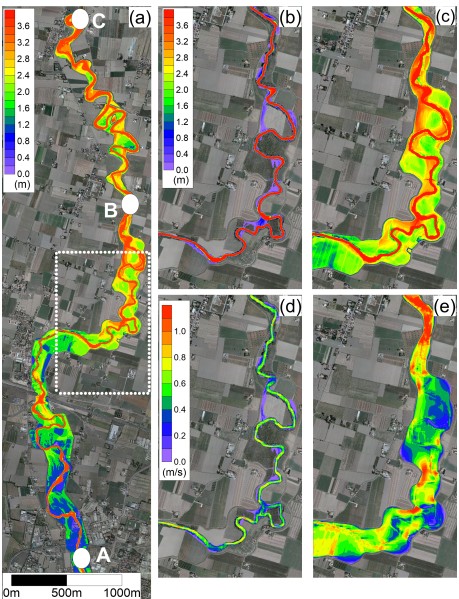

**Figure 7.** Map of the maximum simulated water depths for the Parma River (a): the upstream (A) and downstream (C) boundary conditions and the intermediate observation site (B) are indicated. With reference to the area marked with dotted white line in (a), (b) and (c) represent the water depths and (d) and (e) the velocity field at low and high discharge values, respectively.

were adopted. The Manning roughness coefficient was assumed equal to 0.05 s/m$^{1/3}$. The steady-state values of water depth and velocity fields, obtained considering the initial discharge value of the hydrograph, were adopted as initial conditions.

The inflow condition to be estimated was assumed as follows (D'Oria et al. (2015)):

$$Q(t) = A + B \cdot f(t, b, k), \tag{21}$$

where $t$ denotes the time, $A$ the base flow (constant value), $B$ the volume above the base flow (constant value) and $f$ the gamma distribution, which states:

$$f(t, b, k) = \frac{1}{k^b \Gamma(b)} t^{b-1} e^{-\frac{t}{k}}, \tag{22}$$

where $\Gamma(b)$ represents the gamma function defined through the parameters $b$ and $k$ that denote the shape and the scale parameter, respectively. The parameters of the gamma distribution were set as follows: $A$ = 100 m$^3$/s, $B$ =3·10$^7$ m$^3$, $b$ = 6 and $k$ = 10000 s. The resulted flood wave presented a peak value of about 630 m$^3$/s at time $(b-1)k \approx 14$ hours (Fig. 8-a).

During the estimation, when the sensitivity to the first parameter $p_1$ is investigated, the steady-state flow for the initial discharge is also recomputed. This means that parameter $p_1$ determines not only the first value of the estimated flood wave but also governs the initial condition of the river reach.

The inflow hydrograph duration was limited to 40 hours and it was discretized using 2 hours time interval ($N_p$=21), whereas the observation stage hydrographs were discretized every 0.5 hours. The prior pdf was defined by means of a Gaussian covari-

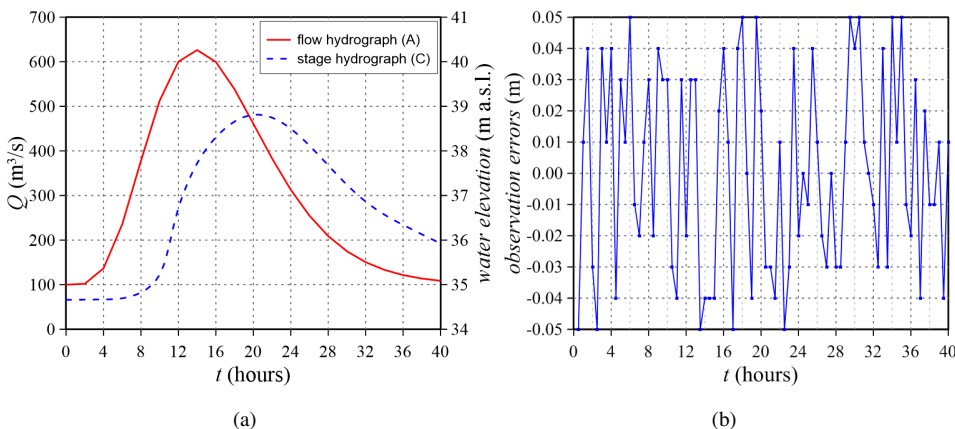

**Figure 8.** Parma River: flow and stage hydrographs at sections A and C, respectively (a) and observation error distribution (b).

ance function, and the initial structural parameters were set as reported in Table 1. In order to avoid non-physical discharge values during the computations, non-negativity was enforced to the unknown parameters by performing the estimation in a logarithmic space. The initial model parameter values were defined by applying the linesearch tool of the bgaPEST, which damps the solution between successive iterations (Fienen et al. (2013)), and avoids numerical instabilities that may occur starting from
a first choice of the parameters too far from the true one.

The inflow hydrograph was estimated first considering true observations (the variance was set equal to $10^{-8}$ m$^2$ to take into account the truncation error). Then, the same discharge hydrograph was defined corrupting the observed water levels with random errors uniformly distributed with maximum deviations of $\pm 0.05$ m and variance $10^{-3}$ m$^2$ (Fig. 8-b).

Qualitative assessment of the inverse methodology is achieved by comparing the reference with the estimated inflow hydro-
graph, as well as the observed with the modeled water levels in the observation site. Considering the simulation without errors in the observations, Fig. 9 shows that the estimated flood wave overlaps the reference one (a), and the modeled water levels agree almost perfectly with the measured ones (b).

The results of the simulation with random errors corrupting the observations are depicted in Fig. 10. The estimated flood wave well matches again the reference one, presenting a misfit relative to the peak value lower than $5\%$, and similarly the
modeled water levels reproduce the reference ones with residual less than $1\%$. Only the last value of the reconstructed flood wave is slightly overestimated, since the more the tested parameter nears the end of the wave, the fewer observations contain information about the related effects, as illustrated by the increasing range of the $95\%$ credibility interval. However, the "true" discharge values are inside the $95\%$ credibility interval, thus confirming the good accuracy of the solution. In addition to this behaviour at the end of the discharge hydrograph (that can be postponed extending the hydrograph total duration), very small
differences between the observed and modelled variables appear when abrupt changes in the inflow function are present (e.g. the initial transition from the steady state to the flood wave). This behaviour is due to the regularization introduced into the solution by the prior information that imposes some degree of continuity and/or smoothness to the estimated hydrograph. However, the

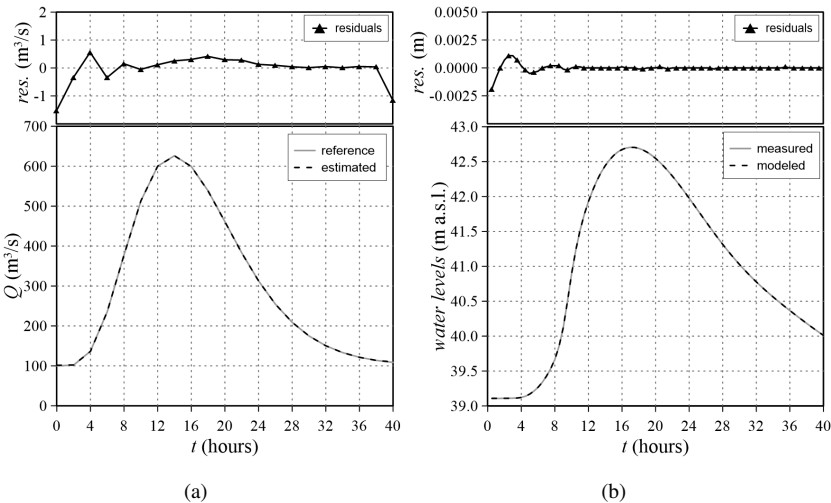

(a)    (b)

**Figure 9.** Parma River: reference and estimated inflow hydrograph (a) and observed (uncorrupted) and modeled water levels (b). The residuals between reference and estimated values are also reported.

residuals are practically negligible and abrupt discontinuities in the inflow hydrographs are not common in natural floods. The

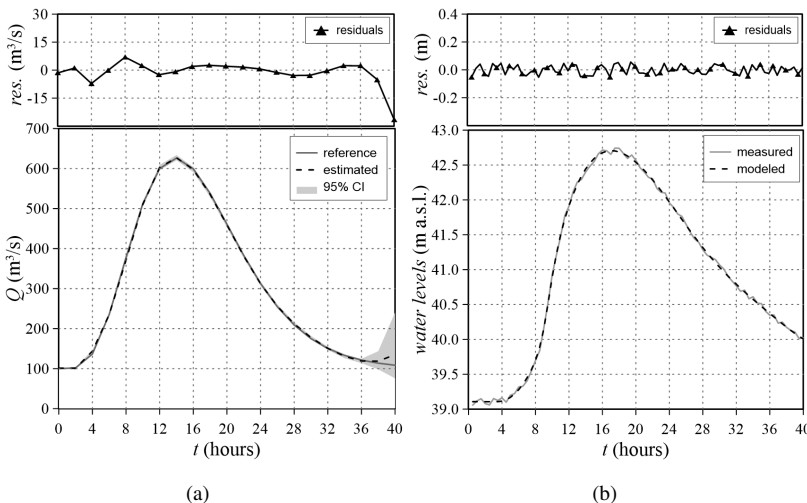

(a)    (b)

**Figure 10.** Parma River: reference and estimated (with 95% credibility interval) inflow hydrograph (a) and observed (corrupted) and modeled water levels (b). The residuals between reference and estimated values are also reported.

structural parameters and the epistemic error variance estimated in the presence and absence of corrupted observations are reported in Table 1.

**Table 1.** Parma River: initial and estimated structural parameters and epistemic error variance.

|  |  | No random errors | Random errors |
|---|---|---|---|
| $\sigma_R^2$ (m$^2$) | Initial | - | 1.00E-4 |
|  | Estimated | - | 1.09E-3 |
| $\sigma_S^2$ (m$^6$s$^{-2}$) | Initial | 5.00E+2 | 5.00E+2 |
|  | Estimated | 1.07E+3 | 5.36E+1 |
| $l$(s) | Initial | 6.48E+4 | 6.48E+4 |
|  | Estimated | 2.90E+4 | 5.28E+4 |

Assessment of the methodology accuracy has been quantified by means of the Nash-Sutcliffe $E_h$, root mean square error RMSE and error in the peak discharge $E_p$ values reported in Table 2. The $E_h$ values are greater than 99%, the $E_p$ values are almost negligible and the RMSE error is less than 0.5 m$^3$/s without random errors and reaches the maximum value of 6 m$^3$/s with corrupted observations.

**Table 2.** Parma River: Nash-Sutcliffe $E_h$, root mean square error RMSE and error in the peak discharge $E_p$ values.

|  | $E_h$ (-) | RMSE (m$^3$/s) | $E_p$ (%) |
|---|---|---|---|
| No random errors | 99.99 | 0.49 | -0.04 |
| Random errors | 99.88 | 6.65 | 0.15 |

## 4.2 Inflow hydrograph estimation on the Secchia River

The second test case concerns both a different river reach and shape of the inflow hydrograph. The studied domain includes a 25 km-long reach of the Secchia River (Northern Italy) between the outflow of the flood control reservoir of Rubiera-Campogalliano located west of Modena town (point A) and the gauging station of Ponte Bacchello (point C) and referring the water level observations to the gauging station of Ponte Alto (point B) (Fig. 11). The modeled river reach is characterized by the presence of many flood plains and floodable areas that influence the flood propagation. The bathymetry was derived from a 1-m resolution DTM obtained through a LiDAR survey carried out in drought condition.

The domain was discretized by means of a non-uniform BUQ grid (Vacondio et al. (2017)), resulting in $77 \cdot 10^3$ computing cells. The Manning roughness coefficient in the riverbed was assumed equal to 0.05 s/m$^{1/3}$ (Vacondio et al. (2016)).

The discharge hydrograph to be estimated is the flood wave of 20-years return period of the Secchia River with a peak value of about 780 m$^3$/s. In order to increase the non-smoothness of the wave, a quite abrupt increment that separates the initial

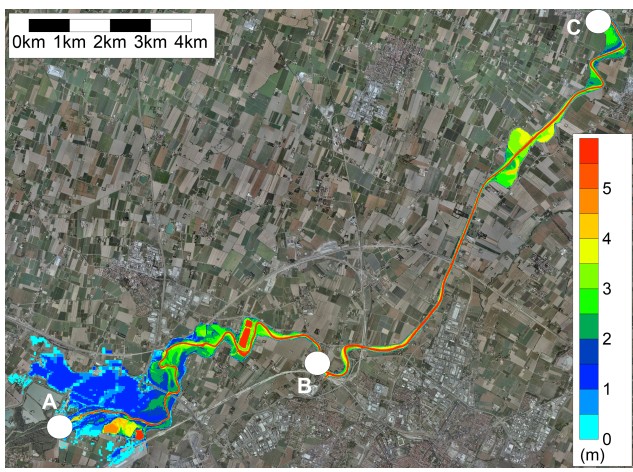

**Figure 11.** Map of the water depths at the flood peak occurrence on the Secchia River, with indication of the upstream (A) and downstream (C) boundary conditions and the intermediate observation site (B).

steady-state condition (100 m³/s) from the rising limb was introduced (Fig. 12-a). It is noteworthy that this flow hydrograph is characterized by a pseudo-real irregular shape, that cannot be properly approximated by an analytical parametric function (e.g., Gamma distribution, Pearson function). The inflow hydrograph ended in 72 hours and was discretized using 2-hours time interval ($N_p$=37), whereas the observed stage hydrograph was discretized every 0.5 hours. The inflow hydrograph was first estimated assuming the true water levels extracted at section B with only a truncation error resulting in a variance of $10^{-8}$ m², and then considering corrupted observations with random errors uniformly distributed with maximum deviations of ±0.05 m and variance $10^{-3}$ m² (Fig. 12-b). Figure 12-a depicts also the discharge hydrograph at the downstream boundary condition section C, in order to highlight the attenuation effect exerted by the flood plains and floodable areas.

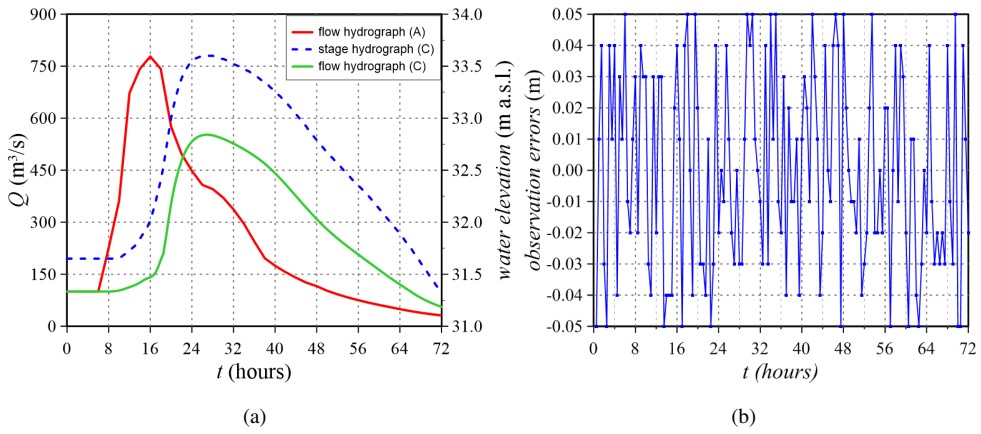

**Figure 12.** Secchia River: flow hydrograph at section A and flow and stage hydrographs at section C (a) and observation error distribution (b).

As before, the parameters were estimated in a logarithmic space and their initial values were calculated adopting the line-search tool of the bgaPEST (Fienen et al. (2013)). The prior pdf was described by means of a linear and Gaussian covariance function, in the configuration with and without corrupted observations, respectively (Table 3).

As shown in Fig. 13 for the simulation without corrupted observations, the estimated flood wave matches almost perfectly the reference one, and the modeled water levels agree with the measured ones.

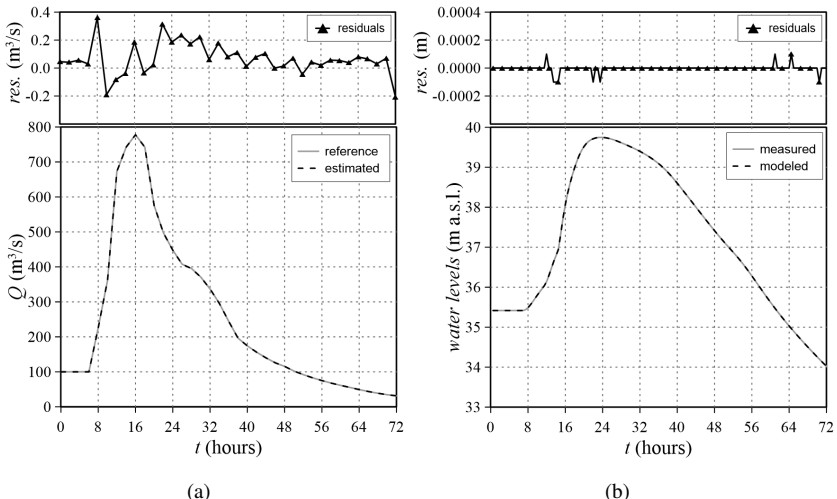

(a)                                                        (b)

**Figure 13.** Secchia River: reference and estimated inflow hydrograph (a) and observed (uncorrupted) and modeled water levels (b). The residuals between reference and estimated values are also reported.

The results of the simulation with corrupted observations depicted in Fig. 14 highlight that both the shape and the peak value are well captured. The small discrepancies of the estimated peak flood wave from the reference one are essentially caused by the fact that the portion with the peak is discretized with only a few parameters and the adopted covariance function smooths the solution.

The structural parameters and the epistemic error variance estimated in the presence and absence of corrupted observations are reported in Table 3.

The indicators used for evaluating the accuracy of the methodology are reported in Table 4. The Nash-Sutcliff efficiency $E_h$ values exceed 99%, the errors in the peak flow $E_p$ are almost negligible and the RMSE error is less than 1 m$^3$/s without random errors and reaches the maximum value of 16 m$^3$/s with corrupted observations: these values highlight the accuracy of

the procedure in estimating the overall shape and peak of the inflow hydrograph.

For this case, some details about the computational characteristics are reported in Table 5.

The computational time of the whole inflow hydrograph simulation (72 hours) is 9.62 minutes, whereas the simulations for evaluating the Jacobian matrix and testing parameters from 2 till 37 required a computational time progressively lower than 9.62 minutes, thanks to the restart option illustrated in the Sect. 3. In order to evaluate the total time required by the

inverse procedure, it is noteworthy that dealing with an HPC cluster the global run time depends on the number of the available

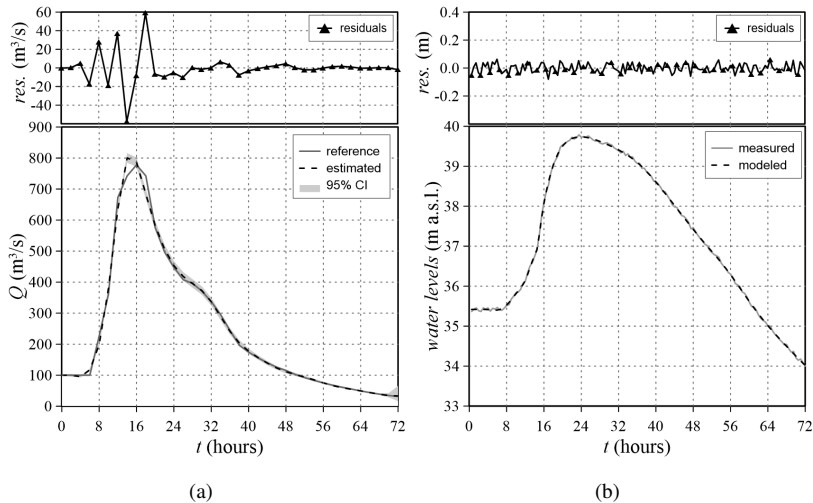

**Figure 14.** Secchia River: reference and estimated (with 95% credibility interval) inflow hydrograph (a) and observed (corrupted) and modeled water levels (b). The residuals between reference and estimated values are also reported.

**Table 3.** Secchia River: initial and estimated structural parameters and epistemic error variance.

|  |  | No random errors | Random errors |
|---|---|---|---|
| $\theta$ (m$^6$s$^{-3}$) | Initial | 1.00E-10 | - |
|  | Estimated | 3.97E-6 | - |
| $\sigma_R^2$ (m$^2$) | Initial | - | 1.00E-4 |
|  | Estimated | - | 1.11E-3 |
| $\sigma_S^2$ (m$^6$s$^{-2}$) | Initial | - | 5.00E+2 |
|  | Estimated | - | 1.38E+1 |
| $l$(s) | Initial | - | 4.32E+4 |
|  | Estimated | - | 3.88E+4 |

GPUs. However, this test was performed using 10 GPUs and the computational cost of the 609 runs was about 13 hours. Since the implemented procedure that manages the interaction between host and server can be used for different HPC cluster, the availability of a cluster equipped with $N_p$ GPUs would have allowed the estimation of the flood wave in about 8 hours. On the other side, the adoption of the serial bgaPEST procedure and the PARFLOOD code as routing model would have required about 4 days of computations that means about 8 times higher than the parallel procedure here proposed. Particularly interesting is the hypothetical evaluation of the computational time for a serial BGA procedure and the adoption of a serial CPU code as forward hydraulic model. Vacondio et al. (2014) pointed out that the PARFLOOD code led to speedup up to two order of magnitude if

**Table 4.** Secchia River: Nash-Sutcliffe $E_h$, root mean square error RMSE and error in the peak discharge $E_p$ values.

|                  | $E_h$ (-) | RMSE (m$^3$/s) | $E_p$ (%) |
| ---------------- | --------- | -------------- | --------- |
| No random errors | 99.99     | 0.13           | -0.02     |
| Random errors    | 99.44     | 16.57          | 2.89      |

**Table 5.** Secchia River: characteristics of the simulation.

| | |
| --- | --- |
| Number of parameters $N_p$ | 37 |
| Physical total time of the inflow hydrograph | 72 hours |
| Physical total time of the run testing the $1^{st}$ parameter $p_1$, assuming 100 hours for reaching the steady state condition | 172 hours |
| Computational time of the whole inflow hydrograph simulation (72 hours) | 9.62 minutes |
| Computational time of the run testing the $1^{st}$ parameter (172 hours) | 19.38 minutes |
| Number of the BGA iterations $N_i$ for the model parameter estimation | 4 |
| Number of the BGA iterations $N_o$ for the structural parameter estimation | 4 |
| Total number of simulations $N_t$ (Eq. 15) | 609 |

compared to a serial CPU code. Therefore, if a serial BGA procedure and the GPU forward model would have required about 4 computational days, the inverse problem solution with a serial forward code would ended in 400 computational days, making the use of the inverse procedure practically infeasible.

## 5   Reconstruction of a historical event: the December 2009 flood wave on the Secchia River

The inverse procedure is now validated by investigating the December 2009 flooding event on the Secchia River, which is one of the most significant events occurred in the last ten years in this river. The Interregional Agency for the Po River (AIPo) monitored the river and provided the water stage hydrographs recorded in the two gauging stations indicated in Fig. 11 with letters B and C, respectively. As shown in Fig. 15, the recorded water levels present more than one rising and recession limb, and thus, besides the challenges related to a real field application, this test aims at addressing also the estimation of an inflow with multiple peaks. In order to estimate the discharge at section A (Fig. 11), the water levels recorded at point B and C were assumed as observations and downstream boundary condition, respectively. The event was simulated from 9 p.m. of the $22^{nd}$ December 2009, till 12 a.m. of the $26^{th}$ December, for a total duration of 87 hours. The water levels were recorded every 0.5 hours, whereas the unknown inflow hydrograph was discretized into 88 parameters (one per hour, $N_p$=88).

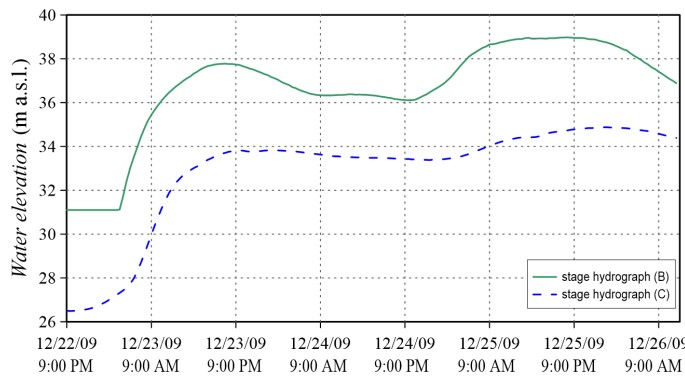

**Figure 15.** December 2009 recorded stage hydrographs on the Secchia River at sections B and C, respectively.

The studied domain is the same previously adopted for a syntethic inflow, and thus, the reader is kindly referred to Sect. 4.2 for the information about bathymetry, initial condition, and roughness configuration.

As before, the parameters were estimated in a logarithmic space and their initial values were calculated adopting the line-search tool of the bgaPEST (Fienen et al. (2013)). The prior pdf was described by means of a Gaussian covariance function; the initial and estimated structural parameters are reported in Table 6.

**Table 6.** Secchia 2009 event: initial and estimated structural parameters.

|  | $\sigma_S^2$ (m$^6$s$^{-2}$) | $l$(s) |
|---|---|---|
| Initial | 5.00E+2 | 6.48E+4 |
| Estimated | 1.49E+1 | 3.36E+4 |

Figure 16 shows the estimated flood wave (and the 95% credibility interval), which presents an irregular shape and two main peaks, as it could be expected from the observed stage hydrograph. Moreover, an additional small intermediate peak is captured that was not so evident from the registered water levels at section B (Fig. 15), even if a little pronounced local maximum can be seen around 3 p.m. of the $24^{th}$ December 2009. The resulting flood wave presents neither instabilities nor oscillations. During the computation, the variance of the epistemic error was assumed equal to $10^{-3}$ m$^2$; as shown in Sect. 4, this means considering the observed water levels corrupted with random errors with maximum deviations of $\pm 0.05$ m. In Fig. 16, the flood wave estimated by increasing the variance of half an order of magnitude is also depicted (dotted line): the solution appears slightly smoothed in a few points, but substantially similar to the inflow resulting with the smaller variance, which is thus considered as the estimated inflow of the studied event. The comparison between modeled and measured water levels at section B is presented in Fig. 17: it is relevant to note that the residuals between the two hydrographs are mostly less than 2 cm and only in a few points of the first rising limb they reach the highest value of 18 cm.

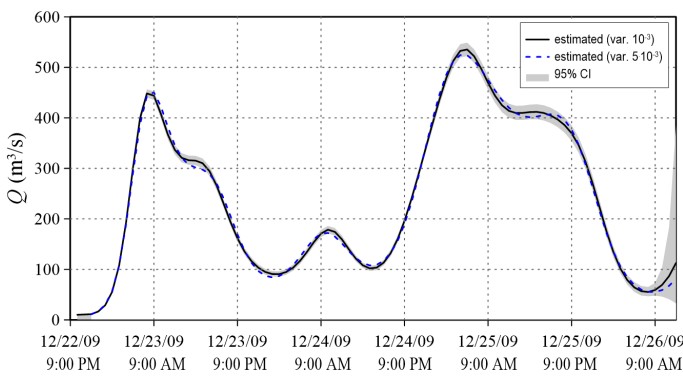

**Figure 16.** Secchia 2009 event: estimated inflow hydrographs assuming the epistemic error variance equal to $10^{-3}$ m$^2$ and $5 \cdot 10^{-3}$ m$^2$, respectively. The 95% credibility interval is referred to the simulation with the epistemic variance equal to $10^{-3}$ m$^2$.

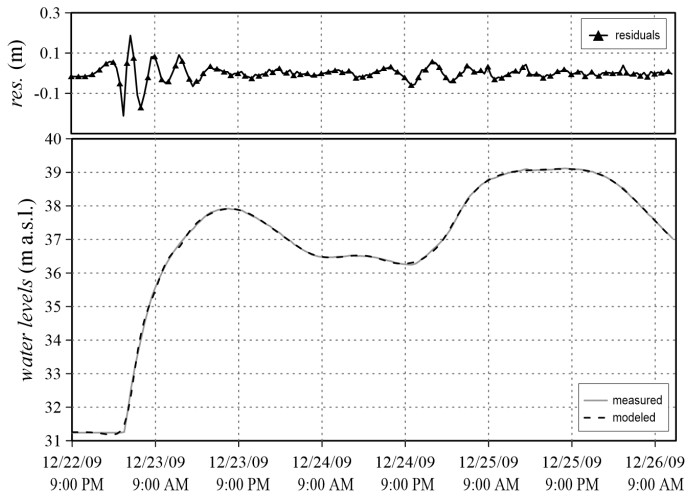

**Figure 17.** Secchia 2009 event: observed and modeled water levels at section B. The residuals between recorded and estimated values are also reported.

With the aim of validating the methodology for this real application, it is noteworthy that the upstream section of the river is located immediately downstream a flood control reservoir equipped with water level sensors. Therefore, the "reference" discharge hydrograph has been obtained from the dam geometrical data (i.e. number and dimension of the bottom openings, crest length of the spillway, etc.) and the recorded water levels adopting the classic hydraulic theory of sluice gates and spillways.

5  Due to the uncertainty in evaluating the discharge coefficients and to the fact that during flood events a large amount of wood debris reduces the outflow discharge from the bottom openings (especially during the depletion phase) and interferes with the overflow spillway, the discharge hydrograph has been calculated adopting equally likely coefficients (Fig. 18). The flood wave estimated by the inverse procedure is in good agreement with the one calculated using the flood reservoir data; the main differ-

ences are after the highest peak, which is well reproduced, although the inverse methodology provides a smoother solution. For this real application, even if the river roughness coefficient was already calibrated in previous studies (Vacondio et al. (2016)), an additional inverse Bayesian estimation was performed with a different value, in order to assess the effect of this coefficient on the solution. Particularly, the Manning coefficient originally set to $0.05$ s/m$^{1/3}$ was decreased by $15\%$ ($0.0425$ s/m$^{1/3}$), as

5 for example can happen due to seasonal changes in vegetation. As shown in Fig. 18, the estimated flood waves are similar and the highest difference, which is in correspondence with the main peak, is less than $6\%$. Therefore, the influence of assuming a "wrong" roughness coefficient is less than linear in the discharge estimation. Despite all the involved approximations, this comparison confirms that the proposed inverse procedure is capable of estimating inflow hydrographs with multiple peaks and irregular shapes in real rivers.

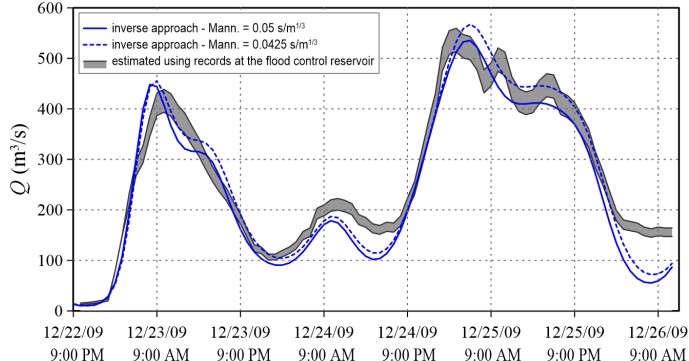

**Figure 18.** Secchia 2009 event: comparison among the inflow hydrographs obtained from the inverse procedure using two different Manning coefficients, and the envelope of different solutions obtained using the records at the flood control reservoir.

## 6    Conclusions

In this work the inverse problem of estimating the unknown inflow hydrograph in an upstream-ungauged section, having water level information only in downstream sites, has been solved by means of a Bayesian methodology. The key aspects in the solution of this problem have been the adoption of a parallel 2D-SWE code running on GPUs and the performance of the sim-

15 ulations over a HPC cluster. The parallelization of the runs useful for the Jacobian matrix computation and the implementation of an *ad hoc* procedure, which allows taking advantage of any HPC cluster with GPUs, have provided a remarkable reduction of the computational costs. For a test case, this parallel procedure reduced the computational time by a factor of 8 against running the 2D-SWE code on a single GPU. Furthermore, the analysis of the runtimes has highlighted that the use of a parallel hydraulic forward routing model is the *conditio sine qua non* for solving this type of inverse problem, whereas the adoption

20 of a serial code would lead to inadmissible computational times. The inverse procedure has been validated considering two different natural rivers; in both tests, no instabilities, due to the adopted inverse procedure or to the availability of a stable, fast

and accurate forward hydraulic model, arose. Moreover, the obtained results have highlighted that the implemented procedure well estimates the unknown inflow hydrographs with different and irregular shapes and in presence of corrupted observations: quantitative indicators have proved the accuracy of the methodology. In all the presented tests, the resulting Nash-Sutcliff efficiency criterion exceeded 99%, the error in the peak discharge was less than 3% and the RMSE error less than 2%. Finally,

5 the proposed inverse procedure allowed the estimation of a historical flood wave characterized by the presence of multiple peaks, without causing instabilities in the solution. The test cases were simulated taking advantage of the HPC cluster of the University of Parma. However, since the implemented procedure is general, it is possible to adopt clouds of GPUs or on-line mini clusters, which are now common and accessible to everyone. The adopted Bayesian software (bgaPEST) is open access and 2D-SWE models are a quite common tools for practitioners, even if till now few of them are fast enough to perform

10 the necessary simulations with a reasonable computing time. Therefore, the 2D coupled methodology here proposed can be adopted in the near future also by professional hydrologists involved for example in the design of hydraulic infrastructures as well as for engineers working on water resource management (i.e. irrigation systems, hydroelectric power stations, etc.) or forensic activities. Future development of the methodology will focus on the possibility of reconstructing the flood waves also in presence of levee breaches and flooding outside the river region, where the adoption of a 2D-SWE model is mandatory.

15 *Competing interests.* The authors declare that they have no conflict of interest.

*Acknowledgements.* This work was partially supported by Ministry of Education, Universities and Research under the Scientific Independence of young Researchers project, grant number RBSI14R1GP, CUP code D92I15000190001. This research benefits from the HPC (High Performance Computing) facility of the University of Parma, Italy. Interregional Agency for the Po River (AIPo) is also gratefully acknowledged for providing data. The authors are grateful to the editor, the anonymous reviewer and Dr. A.D. Koussis for the valuable suggestions

20 on the early version of this manuscript.

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
