# Peer review of "Discharge hydrograph estimation at upstream-ungauged sections by coupling a Bayesian methodology and a 2D GPU Shallow Water model"

_Hydrology and Earth System Sciences, 2018_

## Referee Comment (RC1) · A. D. Koussis (Referee) · 21 May 2018

Estimating an unknown discharge hydrograph at an upstream cross-section is useful in flood hydrology both as a forensic activity (to find the inflow that caused a flood event observed at a certain downstream section) as well as operationally (to determine the operational mode of a reservoir in order to protect a downstream area). Such (rather special, but not rare) problems are tackled either by reverse routing the observed hydrograph to the upstream cross-section (an inverse problem, the solution of which exists, but is not unique and must be regularised; the authors should note, in their relevant section, that the solution does exist), or via optimisation. Both inversion approaches

are subject to instabilities that must be controlled (e.g., smoothing). Past research has been referenced properly.

The submitted research opts for an optimisation approach: the procedure applies a Bayesian geostatistical methodology coupled with forward routing that solves the full 2-D shallow water equations. Using a 2-D flow model in the context of inverse flood routing is an advance beyond the state of the art. But the computational load caused by the necessary multiple 2-D flow runs is heavy. Therefore, the authors have carried out their inversion procedure by parallelising the evaluation of the Jacobian matrix (it assesses the solution sensitivity to each unknown flow value), taking advantage of the floating point calculation capabilities of an array of Graphical Processing Units grouped in a remote High Performance Computing cluster.

The testing and validation of the method is sound and thorough; it includes simulations of generic floods with perfect (error-free) and with corrupted data, as well as of real flood events. The achieved accuracy is very good, including the peak region. Large oscillations of the inverted flow (recovered inflow) hydrograph occurring near its end are explained (Figs. 10 and 16); oscillations occurring at the start of the flood (e.g., Fig. 9a) seem to be due to the somewhat abrupt initiation of the transient from the steady state, while oscillations in the peak region are likely due to the change from a rising to a falling flood flow (Figs. 13a and 14a). The largest oscillations of the stage hydrograph occur at the start of the flood (Figs. 9b and 17) and should be also attributable to the somewhat abrupt initiation of the transient from the steady state (please comment). These oscillations are, of course, stronger in the simulations with corrupted data. Relevant comments by the authors would be appreciated; they would help the reader, too. It is noted, as an aside, that evidence is not conclusive as to which approach, reverse routing or optimisation, is more prone to spurious oscillations; a specific comparative investigation, under identical conditions, is required.

The paper is structured well. The theory is presented succinctly, with adequate mathematics, and contains all relevant information; the same holds for the (important) computational aspects of the modelling approach. Figures and tables add significantly to the understanding of the textual account, and figures are of good quality. The language is generally quite good, yet the paper would benefit from careful editing (e.g., most 'which' should be 'that', 'resulted' should be 'resulting' etc.); some indentations must be corrected. (I will mail my marked up manuscript to the corresponding author for the consideration of the team of authors).

The Conclusions section could be enhanced. Particularly, given that the computing facilities and arrangements required for the inverse modelling approach reported in the manuscript are currently tailored to research rather than to the work of professional hydrologists, the authors should comment on how they envision their model finding its way to the hydrological practice.

Assessment: The paper addresses in a novel way an interesting topic (for specialists) that is within the scope of HESS, is scientifically sound and methodologically solid. It is very good and should be published after minor revision.

---

## Referee Comment (RC2) · Anonymous Referee #2 · 23 May 2018

**General comments**

The manuscript applies a Bayesian geostatistical methodology to the solution of the inverse problem aiming to estimate the upstream flood hydrograph at an un-gauged river section. The downstream routing of the hydrograph is pursued by means of a 2D shallow water model. This leads to a computationally intensive problem, for which a parallel implementation is designed. The most computationally intensive operation (i.e.: the evaluation of the Jacobian matrix) is demanded to a multi-GPU HPC, and also the forward model exploits the opportunities of GPU-parallelization.

The adoption of two-dimensional hydraulic model represents a step forward compared

with both the previous research developed by the Authors and with the state-of-the-art. The resulting complication arising from the increased computational effort is handled properly. Therefore, the research described in the paper appears to be sufficiently innovative, well-designed and of interest to the readers of HESS.

I am rather supportive of the publication of the manuscript, provided that the Authors put some additional effort in improving the quality of the presentation (especially of the English) and in addressing some issues in order to make their outcomes more conclusive. I provide in the following few specific comments to be considered in the revision, as well as some minor issues that could contribute to improve the quality of the manuscript.

**Specific comments**

- I appreciate that the presentation of the Bayesian Geostatistical Approach (BGA) is concise but complete of every detail: however I found it not very clear at some points, detailed below:

  1. The "prior mean" defined in eq. (9) should be better commented, explaining why the $\beta$ vector reduces to "a single value" (do the Authors mean the same value for each parameter?), and why the matrix $X$ reduces to "a single vector of ones".
  2. The separation distance $d$ should be defined explicitly.
  3. I wonder about the opportunity of defining $Q_{ss}$ as $Q_{ss}(\theta)$ since the r.h.s. of eq. (6) does not contain $\theta$.
  4. I could not find the definition of $\xi$ appearing in eq. (9) and eq. (13).
  5. The Authors should better explain what they mean with "a flat solution".

- In the scheme depicting the BGA in figure 3, I could not find the condition corresponding to the parameters convergence, which is claimed in the text. According

to the scheme, the inner cycle terminates only when the maximum number of iterations $N_i$ is reached. The Authors should clarify this point and modify accordingly the manuscript and/or the figure. Assuming that also convergence causes termination, the Authors should explain how did they check the convergence.

- The Authors shoud explain how the credibility intervals may be evaluated based on the results of the BGA algorithm, or at least provide a reference to previous literature.

- About the core of the research described in the manuscript, I am mostly concerned about three issues. They should hopefully be addressed in the revised version of the manuscript.

  1. Since the principal innovation comes from the adoption of a 2D forward hydraulic model, the improvement in terms of the quality of the estimated hydrograph deriving from the use of a more detailed (but also demanding) schematization of the hydraulic process should be explicitly assessed. For instance, how wrong is the estimated hydrograph if one uses a 1D model as the forward routing model in one of the presented examples?

  2. Could the Authors discuss (hopefully with the aid of some additional results) the effects of the resolution of the DEM and/or of the values of the roughness parameters on the estimated hydrograph?

  3. I understand the role of the simulations based on synthetic data-sets, with or without accounting for measure corruption in the validation of the procedure. On the other hand, as far as the "real field application" is concerned, I think that a different test case should have been considered, namely one for which the measured hydrograph was available, in order to compare the estimated with the actual one. This not being the case, the evaluation of the procedure performance cannot go further than the "credibility" (in a statistical sense), and the claims by the Authors in the comments ("This real field

application further confirms the capability of the proposed inverse procedure of estimating irregular inflow hydrographs in real rivers") may sound excessive and not fully supported. Could the Authors take into consideration the addition of such an example?

- English should be carefully revised throughout the entire manuscript to match the standards of scientific communication.

**technical corrections**

- Please refer to eq. (5) and (6) as to linear or Gaussian variogram, just the way you did in section 4.2

- Probably in r.h.s. of eq. (14) a "+" sign is missing. Please check.

- Throughout the manuscript, "non linear" should better read "non-linear"

- Please note that actually the r.h.s. of eq. (12) is not a fraction, therefore referring to "denominator of Eq. (12)" makes sense if you are considering the discrete approximation of the Jacobian.

- The description of fig. 6 and the figure itself refer to four cross-sections along the river: an upstream un-gauged one (A), two intermediate (B and C) where water levels are measured, and a fourth one (D) for downstream boundary condition assignment. However, in the presented examples, only a single intermediate measuring cross section is used, so maybe the description and the figure should be consistently simplified.

---

## Author Comment (AC2) · 29 Jun 2018

**The authors gratefully acknowledge the positive and constructive review of the anonymous Referee. In this document the comments provided by the Referee are reported in italic, whereas the authors' response and indications about the original paper modifications are marked in bold fonts.**

*General comments*
*The manuscript applies a Bayesian geostatistical methodology to the solution of the inverse problem aiming to estimate the upstream flood hydrograph at an un-gauged*

[Figure]

*river section. The downstream routing of the hydrograph is pursued by means of a
2D shallow water model. This leads to a computationally intensive problem, for which
a parallel implementation is designed. The most computationally intensive operation
(i.e.: the evaluation of the Jacobian matrix) is demanded to a multi-GPU HPC, and
also the forward model exploits the opportunities of GPU-parallelization.*

*The adoption of two-dimensional hydraulic model represents a step forward compared
with both the previous research developed by the Authors and with the state-of-the-art.
The resulting complication arising from the increased computational effort is handled
properly. Therefore, the research described in the paper appears to be sufficiently
innovative, well-designed and of interest to the readers of HESS.*

*I am rather supportive of the publication of the manuscript, provided that the Authors
put some additional effort in improving the quality of the presentation (especially of
the English) and in addressing some issues in order to make their outcomes more
conclusive. I provide in the following few specific comments to be considered in the
revision, as well as some minor issues that could contribute to improve the quality of
the manuscript.*

**The authors wish to thank the anonymous Referee for his positive overview
about the manuscript.**

*Specific comments*

- *I appreciate that the presentation of the Bayesian Geostatistical Approach (BGA)
  is concise but complete of every detail: however I found it not very clear at some
  points, detailed below:*

  1. *The "prior mean" defined in eq. (9) should be better commented, explaining
     why the vector $\beta$ reduces to "a single value" (do the Authors mean the same*

*value for each parameter?), and why the matrix **X** reduces to "a single vector of ones".*

**We appreciate this comment and we agree with the Referee that more information about the prior mean is needed to facilitate the readers in figuring out the Bayesian Geostatistical Approach philosophy.**
**As a result, in the revised version of the paper, we will reword the involved paragraph commenting the terms that form the prior mean as follows: "The prior mean is defined as $E[\mathbf{s}] = \mathbf{X}\beta$ where $E$ is the expected value, $\beta$ is the vector of drift coefficients, and X is a known matrix of basis functions.**
**In our case $\beta$ is a single unknown scalar, but different drift coefficients can be used to introduce discontinuities in the stochastic function to be estimated (e.g. when the unknown parameters are likely to form distinct populations). For example, in the context of reverse flow routing problems, multiple values of $\beta$ are adopted if more than one inflow hydrograph must be estimated at the same time (e.g. the inflow on both the upstream branches of a river confluence). The matrix of basis function, X, links each unknown parameter with the corresponding element of $\beta$ and, at the same time, specifies the model of the mean (e.g. constant mean, mean with a trend, etc.); in our case the mean is constant and therefore X is a single vector of ones (Fienen et al. (2008))."**

2. *The separation distance **d** should be defined explicitly.*

**We really appreciate this comment and we acknowledge the potential confusion that arises from the use of the term separation distance.**

This is a legacy from the fact that geostatistics is manly used in estimating spatial parameter fields rather than time functions. We will add in the revised version of the paper that d represents the vector of the separation times between all the parameter pairs: $\mathbf{d}_{i,j} = \mathbf{t}_i - \mathbf{t}_j$ with **i,j**=1,…,**N**$_p$, where **t** represents the time associated with each parameter and **N**$_p$ the total number of unknowns.

3. *I wonder about the opportunity of defining $Q_{ss}$ as $Q_{ss}(\theta)$ since the r.h.s. of eq. (6) does not contain $\theta$.*

   We acknowledge the mistake in the original version of our manuscript. The prior covariance matrix in Eq. (6) is not influenced by the slope parameter $\theta$ but by the variance $\sigma_s^2$ and the integral scale $l$; we will correct $\mathbf{Q}_{ss}(\theta)$ as $\mathbf{Q}_{ss}(\sigma_s^2, l)$.

4. *I could not find the definition of $\xi$ appearing in eq. (9) and eq. (13).*

   We agree with the Referee and the definition of $\xi$, which was missing in the original paper, will be included in the revised text. We will add: "In case a linear relationship between parameters and observations (linear forward model) holds, a computationally efficient method to find the best estimate $\hat{\mathbf{s}}$ of vector s (and $\hat{\beta}$ of $\beta$) is obtained introducing the vector $\xi = (\mathbf{HQ}_{ss}\mathbf{H}^T + \mathbf{R})^{-1}(\mathbf{y} - \mathbf{HX}\hat{\beta})$ and solving the following linear system of equations (Fienen et al. (2009)):"

5. *The Authors should better explain what they mean with "a flat solution".*

   We thank the Referee for this comment and we agree that the term

"flat" should be better explained. For this reason, we completely reworded the sentence making clear what we mean with "a flat solution". In the revised paper the involved sentence will be modified as follows: "The starting values for the structural parameters are assigned so that the variability between contiguous parameters is small (flat solution, with a high degree of correlation); complexity is then introduced during the optimization process if supported by the data. The variance of the epistemic errors is assumed close to the expected one."

- *In the scheme depicting the BGA in figure 3, I could not find the condition corresponding to the parameters convergence, which is claimed in the text. According to the scheme, the inner cycle terminates only when the maximum number of iterations $N_i$ is reached. The Authors should clarify this point and modify accordingly the manuscript and/or the figure. Assuming that also convergence causes termination, the Authors should explain how did they check the convergence.*

The Referee is right. We confirm that both the inner loop to estimate the model parameters and the outer one to estimate the structural parameters iterate until convergence or the assumed maximum number of iterations is reached. Therefore, the $2^{nd}$ (inner > $N_i$) and $3^{rd}$ (outer > $N_o$) decision blocks in Fig. 3 of the manuscript do not only check if the maximum number of iterations is reached, but also verify if convergence is achieved. The flow chart will be corrected in the revised paper. Additionally, we will include the definition about convergence. Recalling that the aim of the inverse procedure is to obtain the vector of the unknown parameters s, as well as to quantify the uncertainty in the estimation, the solution is found maximizing

the posterior pdf or, more conveniently, minimizing its negative logarithm (objective function) (Fienen et al., 2013). The linearization process ends if the maximum number of iterations $N_i$ is reached or if the improvement in the objective function (absolute difference between two successive iterations) is below a user defined value. The structural parameter iteration loop (outer loop) progresses until the maximum number of iterations $N_o$ is reached or the norm of the differences between structural parameter values at consecutive iterations is below a user defined value (Fienen et al., 2013).

- *The Authors should explain how the credibility intervals may be evaluated based on the results of the BGA algorithm, or at least provide a reference to previous literature.*

We really appreciate this suggestion and accordingly we will modify the involved paragraph in the revised manuscript. We clarify that, at the end of the parameter estimation, the linearized uncertainties of the unknowns can be evaluated in terms of the posterior covariance matrix of the estimated parameters, (Fienen at al., 2013). In addition, we will explain that the diagonal elements of this matrix represent the posterior variance ($\sigma^2$) of the estimated parameters and that the 95% credibility interval of the solution is approximately equal to $\pm 2\sigma^2$.

- *About the core of the research described in the manuscript, I am mostly concerned about three issues. They should hopefully be addressed in the revised version of the manuscript.*

  1. *Since the principal innovation comes from the adoption of a 2D forward*

*hydraulic model, the improvement in terms of the quality of the estimated hydrograph deriving from the use of a more detailed (but also demanding) schematization of the hydraulic process should be explicitly assessed. For instance, how wrong is the estimated hydrograph if one uses a 1D model as the forward routing model in one of the presented examples?*

**We really thank the Referee for this comment since it allows us to discuss the motivations that led to enhance the serial Bayesian procedure introduced by D'Oria and Tanda (2012) for 1D cases, to 2D forward models. The choice between 1D and 2D models concerns the classical forward propagation rather than the Bayesian application. In fact, in literature the advantages of 2D-SWEs in comparison with 1D schematizations have been thoroughly discussed (e.g. Costabile et al., 2015), assessing that if river reaches present several floodable areas, meanders and floodplains, as it is typical for lowland streams, only 2D models can properly describe the flood propagation. As shown for example in Fig. 7 of the manuscript, in such rivers the low flow at the beginning of the event follows the meanders and water is contained in the main channel, whereas for high discharge the flow involves the river banks and a continuous mass and momentum exchange occurs between the main channel and the river banks and thus the assumptions of 1D models do not hold.**
**Therefore in our opinion, since the physical phenomena can be only accurately simulated by a 2D numerical scheme, no accurate upstream discharge hydrograph can be obtained by adopting 1D models. Finally, coupling the Bayesian approach with a fast, stable and accurate 2D forward model is the first step for reconstructing the discharge hydrograph during a levee failure and/or overtopping that causes the flooding of the nearest lowlands; the authors are also**

working in this direction that clearly requires the adoption of a 2D model.

2. *Could the Authors discuss (hopefully with the aid of some additional results) the effects of the resolution of the DEM and/or of the values of the roughness parameters on the estimated hydrograph?*

We thank the Referee for this useful comment that allows us to clarify some further aspects of the forward numerical modelling.
The mesh design is an issue related to create an accurate forward model. As for every numerical method that aims at describing a physical phenomenon in a spatial domain, the mesh must be chosen considering both the needed accuracy and the required computational effort. Firstly, the mesh must be defined in such a way that the bathymetry of the rivers is adequately resolved. Figure 1 shows that the adopted mesh (with △x=10 m inside the river) is able to accurately reproduce the river geometry. Secondly, the grid size must guarantee that the numerical solution is close to the "exact solution" of the SWEs. Convergence analysis can be proficiently performed for simple test cases, in which the mesh can be progressively halved many times with a reasonable computational effort. A similar analysis was done in a previous study conducted by some of the present authors (Aureli et al., 2008) and it is beyond the scope of this work. Anyhow, grid size, roughness estimation and numerical discretization of SWEs, all play an interlaced role on the solution results. First order accurate models, for example, intrinsically introduce more dissipation into the solution and this behaviour must be counterbalanced during the calibration phase, for example reducing the dissipation term due to friction, since a part of the dissipation is already embedded in the intrinsic numerical viscosity of the model. Despite the calibration of the considered river (grid size, roughness and numerical discretization) was already assessed in previous studies (Vacondio et al., 2016), according to the Referee' suggestion we performed an additional inverse Bayesian estimation with a different roughness coefficient (please refer also to the next comment answer). Particularly, the Manning coefficient originally set equal to 0.05 s/m$^{1/3}$ was decreased by 15% and assumed equal to 0.0425 s/m$^{1/3}$, as for example can happen due to seasonal changes in vegetation. As shown in Figure 2, the estimated flood waves are similar and the highest difference, which is in correspondence with the second peak, is less than 6%. Therefore, the influence of assuming a "wrong" roughness coefficient is less than linear in the discharge estimation. However, we want to stress that the same issue holds for any model setup. The revised version of the paper will include this analysis concerning the roughness values.

3. *I understand the role of the simulations based on synthetic data-sets, with or without accounting for measure corruption in the validation of the procedure. On the other hand, as far as the "real field application" is concerned, I think that a different test case should have been considered, namely one for which the measured hydrograph was available, in order to compare the estimated with the actual one. This not being the case, the evaluation of the procedure performance cannot go further than the "credibility" (in a statistical sense), and the claims by the Authors in the comments ("This real field application further confirms the capability of the proposed inverse procedure of estimating irregular inflow hydrographs in real rivers") may sound excessive and not fully supported. Could the Authors take into consideration the addition of such an example?*

**We really appreciate this comment and the suggestion pointed out by**

the Referee. As claimed in the paper, the key point of the proposed procedure is to define a discharge hydrograph in an upstream river section that has no records (either in terms of water levels or discharges). However, to validate the methodology for a real field application, a measured reliable discharge hydrograph has to be available in the upstream section of the model. Usually, discharge hydrographs in natural rivers are obtained by converting registered water levels with an appropriate rating curve. Nevertheless, this procedure is affected by two different major uncertainties: (i) rating curves are usually calibrated only for small discharges (ii) the inertial terms of the SWEs, which cause a non-unique level-discharge relationship during floods, are neglected, despite the fact that they are not negligible in lowland rivers. Since in the real field application presented in Sect. 5 the upstream section of the river is located immediately downstream a flood control reservoir dam, the discharge hydrograph has been obtained by adopting the classical hydraulic theory of sluice gates and spillways, partially overcoming the previous issues.

With this aim, we recovered the dam geometrical data from the competent Authority (i.e. number and dimension of the bottom openings, crest length of the spillway, etc.). Please note that during flood events large wood debris accumulates in the reservoir reducing the outflow discharge from the bottom openings, especially during the depletion phase, and disturbing the overflow spillway (see Figure 3). Due to this issue and the uncertainty in evaluating the discharge coefficients, the calculated flood wave showed in Figure 2 has an envelope of different solutions obtained adopting equally likely coefficients.

As depicted in Figure 2, the estimated flood wave is in good agreement with the measured one; the main differences are after the highest peak, which is well reproduced, where the "observed" wave presents two

**small peaks, whereas the inverse methodology provides a smoother solution. Despite all the involved approximations, this comparison confirms that the proposed inverse procedure is capable of estimating inflow hydrographs with multiple peaks and irregular shapes in real rivers. In addition to the flow hydrograph estimated with the Manning roughness coefficient equal to 0.05 s/m$^{1/3}$, Figure 2 reports the inverse estimation with a different roughness coefficient equal to 0.0425 s/m$^{1/3}$ as highlighted in the answer to the second Referee issue. Both these analyses will be included in the revised version of the paper.**

- *English should be carefully revised throughout the entire manuscript to match the standards of scientific communication.*

**We thank the Reviewer for his suggestion. The entire manuscript will be carefully revised. Moreover, the language corrections kindly provided by Dr. A. D. Koussis (first Referee) will be considered in the revised manuscript.**

*technical corrections*

- *Please refer to eq. (5) and (6) as to linear or Gaussian variogram, just the way you did in section 4.2*

**We thank the Reviewer for his technical corrections that we will include in the revised manuscript.**

- *Probably in r.h.s. of eq. (14) a "+" sign is missing. Please check.*

**The Referee is right: we will correct this in the revised paper.**

- *Throughout the manuscript, "non linear" should better read "non-linear"*

**We thank the Reviewer for this suggestion: the correction will be included in the revised manuscript.**

- *Please note that actually the r.h.s. of eq. (12) is not a fraction, therefore referring to "denominator of Eq. (12)" makes sense if you are considering the discrete approximation of the Jacobian.*

**We totally agree with the Referee and, as consequence, we will reformulate the involved paragraph in the revised version of the manuscript. In Sect. 2, where the theory of the Bayesian approach is described, Eq. (12) defines the Jacobian matrix formulation, which is not a fraction but a partial derivative. Therefore, in Sect. 3 of the revised paper we will properly refer to Eq. (12) as the formula to calculate the sensitivity matrix components. Moreover, we will clarify that the Jacobian matrix is approximated according to a finite difference scheme, and hence each element is evaluated as the ratio between the variation of each observation (numerator) for given variation of each parameter (denominator).**

- *The description of fig. 6 and the figure itself refer to four cross-sections along the river: an upstream un-gauged one (A), two intermediate (B and C) where water levels are measured, and a fourth one (D) for downstream boundary condition assignment. However, in the presented examples, only a single intermediate measuring cross section is used, so maybe the description and the figure should be consistently simplified.*

[Figure]

We really thank the Referee for this comment and we acknowledge that the role of the section D was not clear in the manuscript. However, the presence of section D plays a specific role in setting up the synthetic case to use as benchmark for the inverse procedure. We will make it clear in the revised version of the paper adding that: "The information in section D is only preparatory for setting up the synthetic cases and it is not used in the inverse procedure. Imposing a rating curve in D allows to obtain water levels with a non-unique stage-discharge relationship in section C, which is more close to the real circumstances when applying the inverse procedure."

**References**

Aureli, F., Maranzoni, A., Mignosa, P., and Ziveri, C. A weighted surface-depth gradient method for the numerical integration of the 2D shallow water equations with topography. Advances in Water Resources, 31(7), 962-974, 2008.

Costabile, P., Macchione, F., Natale, L., and Petaccia, G. Flood mapping using LIDAR DEM. Limitations of the 1-D modeling highlighted by the 2-D approach. Natural Hazards, 77(1), 181-204, 2015.

D′Oria, M. and Tanda, M. G.: Reverse flow routing in open channels: A Bayesian Geostatistical Approach, Journal of hydrology, 460, 130–135, 2012.

Fienen, M. N., Clemo, T., and Kitanidis, P. K.: An interactive Bayesian geostatistical inverse protocol for hydraulic tomography, Water Resources Research, 44, 2008.

Fienen, M., Hunt, R., Krabbenhoft, D., and Clemo, T.: Obtaining parsimonious hydraulic conductivity fields using head and transport observations: A Bayesian geostatistical parameter estimation approach, Water resources research, 45, 2009.

Fienen, M. N., D'Oria, M., Doherty, J. E., and Hunt, R. J.: Approaches in highly parameterized inversion: bgaPEST, a Bayesian geostatistical approach implementation with PEST: documentation and instructions, Tech. rep., US Geological Survey, 2013.

Vacondio, R., Aureli, F., Ferrari, A., Mignosa, P., Dal Palù, A. Simulation of the January 2014 flood on the Secchia River using a fast and high-resolution 2D parallel shallow-water numerical scheme. Natural Hazards, 80(1), 103-125, 2016.

[Figure]

[Figure]

**Fig. 1.** Ponte Alto section on the Secchia River: comparison between the sections extracted from a 5 m and 10 m resolution DTM.

[Figure]

**Fig. 2.** Secchia 2009 event: comparison among the inflow hydrographs assuming two different Manning coefficients, and the one calculated using the records at the flood control reservoir.

[Figure]

[Figure]

**Fig. 3.** Large wood debris accumulation on the Secchia flood control reservoir during a flood event close to the bottom openings (left) and on the overflow spillway (right).

---

## Author Response (AR1)

Prof. Roberto Greco,

Editor of Hydrology and Earth System Sciences

02/08/2018

**Re: Manuscript hess-2018-118**

Thank you for your comments regarding our manuscript entitled "Discharge hydrograph estimation at upstream-ungauged sections by coupling a Bayesian methodology and a 2D GPU Shallow Water model" by A. Ferrari, M. D'Oria, R. Vacondio, A. Dal Palù, P. Mignosa, and M.G. Tanda. We have now updated the manuscript following the suggestions of the Reviewers after their analysis of the paper.

Please find attached to this letter a description of changes and our replies to each comment along with an updated manuscript.

We hope that all the points raised by the Reviewers have been satisfactorily addressed. We wish to kindly thank the referees for their careful reviews and invaluable comments and hope to hear from you again on the status of the manuscript.

Yours sincerely,

Alessia Ferrari PhD, Research Assistant Department of Engineering and Architecture University of Parma

**RESPONSE TO REFEREE #1: Dr. A. D. Koussis**

The authors gratefully acknowledge the positive and constructive review of Dr. A.D. Koussis. In this document the comments provided by Dr. A.D. Koussis are reported in italic, whereas the authors' response and indications about the original paper modification are marked in **bold** fonts.

Estimating an unknown discharge hydrograph at an upstream cross-section is useful in flood hydrology both as a forensic activity (to find the inflow that caused a flood event observed at a certain downstream section) as well as operationally (to determine the operational mode of a reservoir in order to protect a downstream area). Such (rather special, but not rare) problems are tackled either by reverse routing the observed hydrograph to the upstream cross-section (an inverse problem, the solution of which exists, but is not unique and must be regularised; the authors should note, in their relevant section, that the solution does exist), or via optimisation. Both inversion approaches are subject to instabilities that must be controlled (e.g., smoothing). Past research has been referenced properly.

We thank the Referee for this comment and we agree that the solution of this kind of inverse problems exists. As a consequence, we have reformulated the sentence about the ill-posedness of the inverse problem in the Introduction, clarifying that the challenges we are dealing with are the non-uniqueness and the instabilities of the solution and not its existence, as follows:

"In the literature, this approach is known as reverse flow routing (D'Oria and Tanda (2012)), an illposed inverse problem that presents two main challenges: the solution may be non-unique, and instabilities may arise during the inversion."

The submitted research opts for an optimisation approach: the procedure applies a Bayesian geostatistical methodology coupled with forward routing that solves the full 2-D shallow water equations. Using a 2-D flow model in the context of inverse flood routing is an advance beyond the state of the art. But the computational load caused by the necessary multiple 2-D flow runs is heavy. Therefore, the authors have carried out their inversion procedure by parallelising the evaluation of the Jacobian matrix (it assesses the solution sensitivity to each unknown flow value), taking advantage of the floating point calculation capabilities of an array of Graphical Processing Units grouped in a remote High Performance Computing cluster.

The testing and validation of the method is sound and thorough; it includes simulations of generic floods with perfect (error-free) and with corrupted data, as well as of real flood events. The achieved accuracy is very good, including the peak region. Large oscillations of the inverted flow (recovered inflow) hydrograph occurring near its end are explained (Figs. 10 and 16); oscillations occurring at the start of the flood (e.g., Fig. 9a) seem to be due to the somewhat abrupt initiation of the transient from the steady state, while oscillations in the peak region are likely due to the change from a rising to a falling flood flow (Figs. 13a and 14a). The largest oscillations of the stage hydrograph occur at the start of the flood (Figs. 9b and 17) and should be also attributable to the somewhat abrupt initiation of the transient from the steady state (please

comment). These oscillations are, of course, stronger in the simulations with corrupted data. Relevant comments by the authors would be appreciated; they would help the reader, too.

We thank the Referee for his comment since this is an excellent point to discuss. In the Bayesian Geostatistical Approach, the main mechanism by which soft knowledge about the unknown parameter function is imparted is through the prior information. In our approach, this soft knowledge is intentionally limited to the choice of a parameterized covariance model (the structural parameters, which control the balance between smoothness and misfit, are also estimated during the process) such that significant flexibility is available to the algorithm. Nevertheless, the behaviour raised by the Referee in correspondence of abrupt changes of the inflow hydrograph is due to the regularization imposed by the prior-information. In order to comment and justify this behaviour, in Sect. 4.1 we have added the following sentences:

"In addition to this behaviour at the end of the discharge hydrograph (that can be postponed extending the hydrograph total duration), very small differences between the observed and modelled variables appear when abrupt changes in the inflow function are present (e.g. the initial transition from the steady state to the flood wave). This behaviour is due to the regularization introduced into the solution by the prior information that imposes some degree of continuity and/or smoothness to the estimated hydrograph. However, the residuals are practically negligible and abrupt discontinuities in the inflow hydrographs are not common in natural floods."

It is noted, as an aside, that evidence is not conclusive as to which approach, reverse routing or optimisation, is more prone to spurious oscillations; a specific comparative investigation, under identical conditions, is required.

We really thank the Referee for this comment. Due to the fact that the 2D Shallow Water Equations in their complete and conservative formulation, which govern the motion of the fluid, cannot be inverted, for the 2D applications presented in the paper no comparison between the reverse flow routing technique and the Bayesian one can be performed. However, D'Oria et al. (2012a, 2012b) compared the two strategies for the level pool routing algorithm used to compute the inflow hydrograph in a reservoir. They showed that in presence of corrupted observations, the reverse routing procedure amplifies the errors, whereas the regularization provided by the optimization Bayesian procedure avoids spurious oscillations in the solution.

The paper is structured well. The theory is presented succinctly, with adequate mathematics, and contains all relevant information; the same holds for the (important) computational aspects of the modelling approach. Figures and tables add significantly to the understanding of the textual account, and figures are of good quality. The language is generally quite good, yet the paper would benefit from careful editing (e.g., most 'which' should be 'that', 'resulted' should be 'resulting' etc.); some indentations must be corrected. (I will mail my marked up manuscript to the corresponding author for the consideration of the team of authors).

**The authors wish to thank the Referee for having provided his marked copy of the manuscript: the suggested corrections have been included in the revised paper.**

The Conclusions section could be enhanced. Particularly, given that the computing facilities and arrangements required for the inverse modelling approach reported in the manuscript are currently tailored to research rather than to the work of professional hydrologists, the authors should comment on how they envision their model finding its way to the hydrological practice.

We really appreciate this useful advice since it allows us to better remark the practical aspects of our work. The definition of a discharge hydrograph in an ungauged river section is a relevant issue for professional hydrologists involved for example in the design of hydraulic infrastructures as well as for engineers working on water resource management (i.e. irrigation system, hydroelectric power stations) or forensic activities. With the aim of solving this problem, we propose an application that requires supercomputer and High Performing Computer clusters. These tools are mostly used for University research activities, but they are not only reserved to these environments. In fact, clouds of GPUs or online mini cluster are now common and thus everyone can manage to access these facilities. Moreover, the adopted Bayesian software (bgaPEST) is open access and 2D Shallow Water Equations models are nowadays a quite common tools for practitioners. Therefore, in the concluding section the following sentences have been added:

"The test cases were simulated taking advantage of the HPC cluster of the University of Parma. However, since the implemented procedure is general, it is possible to adopt clouds of GPUs or on-line mini clusters, which are now common and accessible to everyone. The adopted Bayesian software (bgaPEST) is open access and 2D-SWE models are a quite common tools for practitioners, even if till now few of them are fast enough to perform the necessary simulations with a reasonable computing time. Therefore, the 2D coupled methodology here proposed can be adopted in the near future also by professional hydrologists involved for example in the design of hydraulic infrastructures as well as for engineers working on water resource management (i.e. irrigation systems, hydroelectric power stations, etc.) or forensic activities."

Assessment: The paper addresses in a novel way an interesting topic (for specialists) that is within the scope of HESS, is scientifically sound and methodologically solid. It is very good and should be published after minor revision.

The authors wish to thank Dr. A.D. Koussis for his suggestions and considerations.

**RESPONSE TO REFEREE #2:**

The authors gratefully acknowledge the positive and constructive review of the anonymous Referee. In this document the comments provided by the Referee are reported in italic, whereas the authors' response and indications about the original paper modification are marked in **bold** fonts.

**General comments**

The manuscript applies a Bayesian geostatistical methodology to the solution of the inverse problem aiming to estimate the upstream flood hydrograph at an un-gauged river section. The downstream routing of the hydrograph is pursued by means of a 2D shallow water model. This leads to a computationally intensive problem, for which a parallel implementation is designed. The most computationally intensive operation (i.e.: the evaluation of the Jacobian matrix) is demanded to a multi-GPU HPC, and also the forward model exploits the opportunities of GPU-parallelization.

The adoption of two-dimensional hydraulic model represents a step forward compared with both the previous research developed by the Authors and with the state-of-the-art. The resulting complication arising from the increased computational effort is handled properly. Therefore, the research described in the paper appears to be sufficiently innovative, well-designed and of interest to the readers of HESS.

I am rather supportive of the publication of the manuscript, provided that the Authors put some additional effort in improving the quality of the presentation (especially of the English) and in addressing some issues in order to make their outcomes more conclusive. I provide in the following few specific comments to be considered in the revision, as well as some minor issues that could contribute to improve the quality of the manuscript.

**The authors wish to thank the anonymous Referee for his positive overview about the manuscript.**

**Specific comments**

• I appreciate that the presentation of the Bayesian Geostatistical Approach (BGA) is concise but complete of every detail: however I found it not very clear at some points, detailed below:

1. The "prior mean" defined in eq. (9) should be better commented, explaining why the vector reduces to "a single value" (do the Authors mean the same value for each parameter?), and why the matrix *X* reduces to "a single vector of ones".

We appreciate this comment and we agree with the Referee that more information about the prior mean is needed to facilitate the readers in figuring out the Bayesian Geostatistical Approach philosophy. As a result, at Sect. 2.1.2 of the revised paper, the involved paragraph commenting the terms that form the prior mean has been reword as follows:

"The prior mean is defined as:  $E[s] = X\beta$ , where *E* is the expected value,  $\beta$  is the vector of drift coefficients, and X is a known matrix of basis functions. In our case  $\beta$  is a single unknown scalar, but different drift coefficients can be used to introduce discontinuities in the stochastic function to be estimated (e.g. when the unknown parameters are likely to form distinct populations). For example, in the context of reverse flow routing problems, multiple values of  $\beta$  are adopted if more than one inflow hydrograph must be estimated at the same time (e.g. the inflow on both the upstream branches of a river confluence). The matrix of basis function, X, links each unknown parameter with the corresponding element of  $\beta$  and, at the same time, specifies the model of the mean (e.g. constant mean, mean with a trend, etc.); in our case the mean is constant and therefore X is a single vector of ones (Fienen et al., 2008)."

**2. The separation distance d should be defined explicitly.**

We really appreciate this comment and we acknowledge the potential confusion that arises from the use of the term separation distance. This is a legacy from the fact that geostatistics is manly used in estimating spatial parameter fields rather than time functions. In Sect. 2.1.2 of the revised paper we have explicitly defined the variable as follows:

"... *d* represents the vector of the separation times between all the parameter pairs  $(d_{i,j} = t_i - t_j$  with  $i, j = 1, ..., N_p, t$  denoting the time associated with each parameter and  $N_p$  the total number of unknowns)."

**3. I wonder about the opportunity of defining $Q_{ss}$ as $Q_{ss}(\theta)$ since the r.h.s. of eq. (6) does not contain $\theta$ .**

We acknowledge the mistake in the original version of our manuscript. The prior covariance matrix in Eq. (6) is not influenced by the slope parameter  $\theta$  but by the variance  $\sigma_s^2$  and the integral scale *l*; we have corrected  $Q_{ss}(\theta)$  as  $Q_{ss}(\sigma_s^2, l)$ .

**4. I could not find the definition of $\xi$ appearing in eq. (9) and eq. (13).**

We agree with the Referee and the definition of  $\xi$ , which was missing in the original paper, has been included at Sect. 2.1.3 as follows:

"In case a linear relationship between parameters and observations (linear forward model) holds, a computationally efficient method to find the best estimate  $\hat{s}$  of vector s (and  $\hat{\beta}$  of  $\beta$ ) is obtained introducing the vector  $\xi = (HQ_{ss}H^T + R)^{-1}(y - HX\hat{\beta})$  and solving the following linear system of equations (Fienen et al. (2009)):"

5. The Authors should better explain what they mean with "a flat solution".

We thank the Referee for this comment and we agree that the term "flat" should be better explained. For this reason, we completely reworded the sentence making clear what we mean with "a flat solution" as follows:

"... the starting values for the structural parameters are assigned so that the variability between contiguous parameters is small (flat solution, with a high degree of correlation); complexity is then introduced during the optimization process if supported by the data. The variance of the epistemic errors is assumed close to the expected one."

• In the scheme depicting the BGA in figure 3, I could not find the condition corresponding to the parameters convergence, which is claimed in the text. According to the scheme, the inner cycle terminates only when the maximum number of iterations Ni is reached. The Authors should clarify this point and modify accordingly the manuscript and/or the figure. Assuming that also convergence causes termination, the Authors should explain how did they check the convergence.

The Referee is right. We confirm that both the inner loop to estimate the model parameter and the outer one to estimate the structural parameters iterate until convergence or the assumed maximum number of iterations is reached. Therefore, the  $2^{nd}$  (inner >  $N_i$ ) and  $3^{rd}$  (outer >  $N_o$ ) decision blocks in Fig. 3 of the manuscript do not only check if the maximum number of iterations is reached, but also verify if convergence is achieved. The flow chart in Fig. 3 has been corrected. Additionally, we have included in Sect. 2.1.3 the definition about convergence, as follows:

"Recalling that the aim of the inverse procedure is to obtain the vector of the unknown parameters s, as well as to quantify the uncertainty in the estimation, the solution is found by maximizing the posterior pdf or, more conveniently, minimizing its negative logarithm (objective function) (Fienen et al., 2013)."

"The linearization process ends if the improvement (absolute difference between two successive iterations) in the objective function is below a user defined value or if the maximum number of iterations  $N_i$  is reached. The structural parameter iteration loop (outer loop) progresses until the L2-norm of the differences between structural parameter values at consecutive iterations is below a user defined value or if the maximum number of iterations  $N_o$  is reached (Fienen et al. 2013)."

• The Authors should explain how the credibility intervals may be evaluated based on the results of the BGA algorithm, or at least provide a reference to previous literature.

We really appreciate this suggestion and accordingly we have modified the involved paragraph at the end of Sect. 2.1.3 as follows:

"The diagonal elements of this matrix represent the posterior variance ( $\sigma^2$ ) of the estimated parameters and thus, the 95% credibility interval of the solution is evaluated as  $\pm 2\sigma^2$ ."

• About the core of the research described in the manuscript, I am mostly concerned about three issues. They should hopefully be addressed in the revised version of the manuscript.

1. Since the principal innovation comes from the adoption of a 2D forward hydraulic model, the improvement in terms of the quality of the estimated hydrograph deriving from the use of a more detailed (but also demanding) schematization of the hydraulic process should be explicitly assessed. For instance, how wrong is the estimated hydrograph if one uses a 1D model as the forward routing model in one of the presented examples?

We really thank the Referee for this comment since it allows us to discuss the motivations that led to enhance the serial Bayesian procedure introduced by D'Oria and Tanda (2012) for 1D cases, to 2D forward models. The choice between 1D and 2D models concerns the classical forward propagation rather than the Bayesian application. In fact, in literature the advantages of 2D-SWEs in comparison with 1D schematizations have been thoroughly discussed (e.g. Costabile et al., 2015), assessing that if river reaches present several floodable areas, meanders and floodplains, as it is typical for lowland streams, only 2D models can properly describe the flood propagation. As shown for example in Fig. 7 of the manuscript, in such rivers the low flow at the beginning of the event follows the meanders and water is contained in the main channel, whereas for high discharge the flow involves the river banks and a continuous mass and momentum exchange occurs between the main channel and the river banks and thus the assumptions of 1D models do not hold.

Therefore in our opinion, since the physical phenomena can be only accurately simulated by a 2D numerical scheme, no accurate upstream discharge hydrograph can be obtained by adopting 1D models. Finally, coupling the Bayesian approach with a fast, stable and accurate 2D forward model is the first step for reconstructing the discharge hydrograph during a levee failure and/or overtopping that causes the flooding of the nearest lowlands; the authors are also working in this direction that clearly requires the adoption of a 2D model.

2. Could the Authors discuss (hopefully with the aid of some additional results) the effects of the resolution of the DEM and/or of the values of the roughness parameters on the estimated hydrograph?

We thank the Referee for this useful comment that allows us to clarify some further aspects of the forward numerical modelling.

The mesh design is an issue related to create an accurate forward model. As for every numerical method that aims at describing a physical phenomenon in a spatial domain, the mesh must be chosen considering both the needed accuracy and the required computational effort. Firstly, the mesh must be defined in such a way that the bathymetry of the rivers is adequately resolved. Figure 1 below shows that the adopted mesh (with  $\Delta x=10$  m inside the river) is able to accurately reproduce the river geometry.

Figure 1. Ponte Alto section on the Secchia River: comparison between the sections extracted from a 5 m and 10 m resolution DTM.

Secondly, the grid size must guarantee that the numerical solution is close to the "exact solution" of the SWEs. Convergence analysis can be proficiently performed for simple test cases, in which the mesh can be progressively halved many times with a reasonable computational effort. A similar analysis was done in a previous study conducted by some of the present authors (Aureli et al., 2008) and it is beyond the scope of this work. Anyhow, grid size, roughness estimation and numerical discretization of SWEs, all play an interlaced role on the solution results. First order accurate models, for example, intrinsically introduce more dissipation into the solution and this behaviour must be counterbalanced during the calibration phase, for example reducing the dissipation term due to friction, since a part of the dissipation is already embedded in the intrinsic numerical viscosity of the model. Despite the calibration of the considered river (grid size, roughness and numerical discretization) was already assessed in previous studies (Vacondio et al., 2016), according to the Referee' suggestion we performed an additional inverse Bayesian estimation with a different roughness coefficient for the real field test case in Sect. 5 (please refer also to the next comment answer). Particularly, the Manning coefficient originally set equal to  $0.05 \text{ s/m}^{1/3}$  was decreased by 15% and assumed equal to 0.0425 s/m1/3, as for example can happen due to seasonal changes in vegetation. As shown in Fig. 18 of the revised paper, the estimated flood waves are similar and the highest difference, which is in correspondence with the second peak, is less than 6%. Therefore, the influence of assuming a "wrong" roughness coefficient is less than linear in the discharge estimation. However, we want to stress that the same issue holds for any model setup.

3. I understand the role of the simulations based on synthetic data-sets, with or without accounting for measure corruption in the validation of the procedure. On the other hand, as far as the "real field application" is concerned, I think that a different test case should have been considered, namely one for which the measured hydrograph was available, in order to compare the estimated with the actual one. This not being the case, the evaluation of the procedure performance cannot go further than the "credibility" (in a statistical sense), and the claims by the Authors in the comments ("This real field

application further confirms the capability of the proposed inverse procedure of estimating irregular inflow hydrographs in real rivers") may sound excessive and not fully supported. Could the Authors take into consideration the addition of such an example?

We really appreciate this comment and the suggestion pointed out by the Referee. Since the upstream section A is located immediately downstream a flood control reservoir equipped with water level sensors, we have adopted the classic hydraulic theory of sluice gates and spillways to calculate the "reference" solution. As a result, Sect. 5 has been improved by validating the inflow hydrograph resulted from the inverse procedure as follows:

"With the aim of validating the methodology for this real application, it is noteworthy that the upstream section of the river is located immediately downstream a flood control reservoir equipped with water level sensors. Therefore, the "reference" discharge hydrograph has been obtained from the dam geometrical data (i.e. number and dimension of the bottom openings, crest length of the spillway, etc.) and the recorded water levels adopting the classic hydraulic theory of sluice gates and spillways. Due to the uncertainty in evaluating the discharge coefficients and to the fact that during flood events a large amount of wood debris reduces the outflow discharge from the bottom openings (especially during the depletion phase) and interferes with the overflow spillway, the discharge hydrograph has been calculated adopting equally likely coefficients (Fig. 18). The flood wave estimated by the inverse procedure is in good agreement with the one calculated using the flood reservoir data; the main differences are after the highest peak, which is well reproduced, although the inverse methodology provides a smoother solution. For this real application, even if the river roughness coefficient was already calibrated in previous studies (Vacondio et al. (2016)), an additional inverse Bayesian estimation was performed with a different value, in order to assess the effect of this coefficient on the solution. Particularly, the Manning coefficient originally set to 0.05 s/m1/3 was decreased by 15% (0.0425 s/m1/3), as for example can happen due to seasonal changes in vegetation. As shown in Fig. 18, the estimated flood waves are similar and the highest difference, which is in correspondence with the main peak, is less than 6%. Therefore, the influence of assuming a "wrong" roughness coefficient is less than linear in the discharge estimation."

• English should be carefully revised throughout the entire manuscript to match the standards of scientific communication.

We thank the Reviewer for his suggestion. The entire manuscript has been carefully revised and the language corrections kindly provided by Dr. A. D. Koussis (first Referee) have been integrated in the revised manuscript.

**technical corrections**

• Please refer to eq. (5) and (6) as to linear or Gaussian variogram, just the way you did in section 4.2

We thank the Reviewer for his technical corrections. The formula reported in eq. (5) and (6) express the linear and Gaussian covariance function, respectively, and not the variogram. Accordingly, at Sect. 4.2 of the revised paper we have referred to covariance functions and not variogram.

• Probably in r.h.s. of eq. (14) a "+" sign is missing. Please check.

The Referee is right: this has been corrected in the revised paper.

• Throughout the manuscript, "non linear" should better read "non-linear"

We thank the Reviewer for this suggestion: this has been corrected in the revised manuscript.

• Please note that actually the r.h.s. of eq. (12) is not a fraction, therefore referring to "denominator of Eq. (12)" makes sense if you are considering the discrete approximation of the Jacobian.

We totally agree with the Referee and, as consequence, we have reformulated the involved paragraph in the revised version of the manuscript. In Sect. 2, where the theory of the Bayesian approach is described, Eq. (12) defines the Jacobian matrix computation, which is not a fraction but a partial derivative. Therefore, in Sect. 3 we have reworded the paragraph as follows:

"The simulation of a base run, once a particular set of parameters has been assumed (deriving from the initialization or from previous estimation steps), represents a mandatory step for the Jacobian matrix evaluation, which is performed at this point of the procedure in order to quantify how each observation is influenced by the variation of each estimable parameter. Particularly, Eq. (12) is approximated using a finite difference method, and hence each element of the matrix is evaluated as the ratio between the variation of each observation (numerator) for given variation of each parameter (denominator) with respect to the base run."

• The description of fig. 6 and the figure itself refer to four cross-sections along the river: an upstream ungauged one (A), two intermediate (B and C) where water levels are measured, and a fourth one (D) for downstream boundary condition assignment. However, in the presented examples, only a single intermediate measuring cross section is used, so maybe the description and the figure should be consistently simplified.

We really thank the Referee for this comment and we acknowledge that the role of the section D was not clear in the manuscript. However, the presence of section D plays a specific role in setting up the synthetic case to use as benchmark for the inverse procedure. In the revised manuscript the following sentences have been added:

"The information in sub-reach C-D is only preparatory for setting up the synthetic cases and it is not used in the inverse procedure. Imposing a rating curve in D allows to obtain water levels with a nonunique stage-discharge relationship in section C, which is more close to the real circumstances when applying the inverse procedure."

**Abstract.** In this This paper presents a novel methodology for estimating the unknown discharge hydrograph at the entrance of a river reach, where when no information is available, is presented. The methodology is obtained by coupling couples an optimization procedure, based on the Bayesian Geostatistical Approach (BGA), with a forward self-developed 2D hydraulic model of the stream. In order to accurately describe the flow propagation in real rivers characterized by large floodable areas,

[revised manuscript text omitted]

$$p(\mathbf{s}|\mathbf{y}) \propto L\left(\mathbf{y}|\mathbf{s}\right) p\left(\mathbf{s}\right),\tag{1}$$

where s is the vector of the unknown parameters, y is the vector of the measured data,  $p(\mathbf{s}|\mathbf{y})$  is the posterior probability density function (pdf) of s given y,  $L(\mathbf{y}|\mathbf{s})$  is the likelihood function and  $p(\mathbf{s})$  is the prior probability density function of s. Since the

- 10 present work aims at estimating an upstream hydrograph in an ungauged section, assuming the knowledge of downstream water levels, s represents the discharge values over time of the unknown inflow hydrograph, whereas y denotes the downstream water level observations. Following Eq.(1), the posterior pdf , which represents the parameter knowledge after the observations, can be seen as a combination between a priori knowledge on the parameters (prior pdf), where a priori means that the observed data are still not considered, and information about parameters contained in the measured data (likelihood function) (Glickman and
- 15 Van Dyk (2007)). In the BGA method proposed by Kitanidis (1995), the prior pdf and the likelihood function are described by means of Gaussian distributions and the best set of parameter **s** is obtained by maximizing the posterior pdf.

**2.1.1 The likelihood function**

Focusing on the terms of the Bayes theorem The likelihood function  $L(\mathbf{y}|\mathbf{s})$  in Eq. (1) characterizes the misfit deviation between observed data and model results (Fienen et al. (2013)). Starting from the results of the forward model,  $L(\mathbf{y}|\mathbf{s})$  delineates how a particular set of parameters  $\mathbf{s}$  is able to reproduce the observations  $\mathbf{y}$  in space and/or time, therefore thus accounting for the epistemic errors. The investigated inverse problem presents different sources of errors that , which are related to the conceptual schematization of the inverse procedure, to the numerical forward model and to the data measurement. In the likelihood function, the errors are assumed to be independent and identically distributed, with null zero mean and covariance matrix expressed as follows:

$$\mathbf{25} \quad \mathbf{R} = \sigma_R^2 \mathbf{I},\tag{2}$$

where  $\sigma_R^2$  denotes the variance that regulates expresses the misfit between observed and modeled data, and I is the identity matrix.

**2.1.2 The prior probability density function**

The prior knowledge about  $\mathbf{s}$  (p( $\mathbf{s}$ ) in Eq.(1)) is limited to the definition assignment of a mean value (unknown and estimated 30 during the procedure) and a characteristic about the continuity and/or smoothness of the parameter field described through a covariance function implemented as covariance matrix. It furnishes a soft knowledge about the structure/shape of the unknowns and provides a regularization of the solution; the prior pdf can also be used to enforce non-negativity to the parameters (D'Oria and Tanda (2012)). The prior mean is defined as:

$$E[\mathbf{s}] = \mathbf{X}\boldsymbol{\beta},\tag{3}$$

- 5 where *E* is the expected value,  $\beta$  is the vector of drift coefficients, and **X** is a known matrix of basis functions. In our case  $\beta$  is a single unknown scalar, but different drift coefficients can be used to introduce discontinuities in the stochastic function to be estimated (e.g. when the unknown parameters are likely to form distinct populations). For example, in the context of reverse flow routing problems, multiple values of  $\beta$  are adopted if more than one inflow hydrograph must be estimated at the same time (e.g. the inflow on both the upstream branches of a river confluence). The matrix of basis function, **X**, links each unknown
- 10 parameter with the corresponding element of  $\beta$  and, at the same time, specifies the model of the mean (e.g. constant mean, mean with a trend, etc.); in our case the mean is constant and therefore **X** is a single vector of ones (Fienen et al. (2008)). (a single vector of ones, in this case), which link each value of s with the appropriate element of  $\beta$ .

The prior covariance matrix of the unknown parameters  $\mathbf{Q}_{ss}$  is then defined as:

$$\mathbf{Q}_{ss} = E\left[\left(\mathbf{s} - \mathbf{X}\boldsymbol{\beta}\right)\left(\mathbf{s} - \mathbf{X}\boldsymbol{\beta}\right)^{T}\right].$$
(4)

15 In the context of geostatistics, the covariance matrix  $\mathbf{Q}_{ss}$  is a function of the separation distance (in time in this case) between the parameters and describes their deviations from the mean behavior. Different functions models can be adopted to describe the covariance; for example, it can be assumed as a linear function, represented through a limiting case of the exponential covariance function model (Fienen et al. (2008)), according to the following relation:

$$\mathbf{Q}_{ss}(\theta) = \theta l \exp\left(-\frac{|\mathbf{d}|}{l}\right),\tag{5}$$

20 where **d** represents the vector of the separation distances in time between the parameters, times between all the parameter pairs  $(d_{i,j} = t_i - t_j \text{ with } i,j=1,...,N_p, t \text{ denoting the time associated with each parameter and } N_p$  the total number of unknowns), l a fixed integral scale  $(l = 10 \max(d))$  and  $\theta$  the slope (structural parameter), which influences that governs the correlation between the discharge values of the unknown hydrograph. A different formulation (D'Oria et al. (2014)) defines the prior covariance matrix  $\mathbf{Q}_{ss}$  by means of a Gaussian function model characterized by two structural parameters ( $\sigma_s^2$  and l):

25
$$\mathbf{Q}_{ss}(\sigma_s^2, l) = \sigma_s^2 \exp\left(-\frac{|\mathbf{d}^2|}{l^2}\right),$$
 (6)

where  $\sigma_s^2$  denotes the variance. The linear function (Eq.(5)) enforces only continuity to the solution whereas the Gaussian function model (Eq.(6)) adds also some degree of smoothness, but the final solution is still driven by the observations.

**2.1.3 The posterior probability density function**

With the assumptions made, the likelihood and prior terms that compose the posterior pdf of Eq.(1) can be rewritten as follows (Fienen et al. (2009); D'Oria and Tanda (2012); D'Oria et al. (2014)):

$$L(\mathbf{y}|\mathbf{s}) = \exp\left(-\frac{1}{2}\left(\mathbf{y} - \mathbf{h}(\mathbf{s})\right)^T \mathbf{R}^{-1}\left(\mathbf{y} - \mathbf{h}(\mathbf{s})\right)\right)$$
(7)

5

$$p(\mathbf{s}) = \exp\left(-\frac{1}{2}\left(\mathbf{s} - \mathbf{X}\boldsymbol{\beta}\right)^T \mathbf{Q}_{ss}^{-1}\left(\mathbf{s} - \mathbf{X}\boldsymbol{\beta}\right)\right)\right).$$
(8)

The term h(s), in the likelihood function, represents the modeled values in the same place and time of as the available observations v. Therefore, to evaluate h(s), a forward model of the considered river reach that is able to describe the hydraulic routing process is required in order to provide, for a given set of parameter s, the corresponding downstream water levels.

10

Recalling that the aim of the inverse procedure is to obtain the vector of the unknown parameters s, as well as to quantify the uncertainty in the estimation, the solution is found by maximizing the posterior pdf or, more conveniently, minimizing its negative logarithm (objective function) (Fienen et al. (2013)).

In case a linear relationship between parameters and observations (linear forward model) holds, a computationally efficient method to find the best estimate  $\hat{\mathbf{s}}$  of vector  $\mathbf{s}$  (and  $\hat{\boldsymbol{\beta}}$  of  $\boldsymbol{\beta}$ ) is obtained introducing the vector  $\boldsymbol{\xi} = (\mathbf{H}\mathbf{Q}_{ss}\mathbf{H}^T + \mathbf{R})^{-1}(\mathbf{y} - \mathbf{H}\mathbf{X}\hat{\boldsymbol{\beta}})$

and solving the following linear system of equations (Fienen et al. (2009)): In case a linear relationship between parameters 15 and observations (linear forward model) holds, the best estimate  $\hat{s}$  of vector s (and  $\hat{\beta}$  of  $\beta$ ) is obtained by solving the following linear system of equations

$$\begin{cases} \hat{\mathbf{s}} = \mathbf{X}\hat{\boldsymbol{\beta}} + \mathbf{Q}_{ss}\mathbf{H}^{T}\boldsymbol{\xi} \\ \begin{bmatrix} \mathbf{H}\mathbf{Q}_{ss}\mathbf{H}^{T} + \mathbf{R} & \mathbf{H}\mathbf{X} \\ \mathbf{X}^{T}\mathbf{H}^{T} & 0 \end{bmatrix} \begin{bmatrix} \boldsymbol{\xi} \\ \hat{\boldsymbol{\beta}} \end{bmatrix} = \begin{bmatrix} \mathbf{y} \\ 0 \end{bmatrix}, \tag{9}$$

where **H** is the sensitivity (Jacobian) matrix, representing how the observations y are influenced by  $\frac{a \text{ single each}}{a \text{ single each}}$  unknown parameter  $s_i$  (D'Oria et al. (2015)). However, for this particular problem the particular problem under investigation, h(s) is 20 non-linear and therefore matrix **H** depends on **s**. Following the quasi-linear geostatistical approach (Kitanidis (1995)), the relationship between observations and parameters can be successively linearized about a candidate solution  $\mathbf{s}_k$ , where k represents each iteration for each iteration k, the relationship between observations and parameters is successively linearized about a candidate solution:

25
$$\mathbf{h}(\mathbf{s}) \approx \mathbf{h}(\mathbf{s}_k) + \mathbf{H}_k(\mathbf{s} - \mathbf{s}_k),$$
 (10)

and then a correction to the measurements is applied according to the following relation:

$$\mathbf{y}_k = \mathbf{y} - \mathbf{h}(\mathbf{s}_k) + \widetilde{\mathbf{H}}_k \mathbf{s}_k. \tag{11}$$

Therefore, the sensitivity matrix is evaluated at each iteration as follows (D'Oria et al. (2014)):

$$\widetilde{\mathbf{H}}_{k} = \left. \frac{\partial \mathbf{h}(\mathbf{s})}{\partial \mathbf{s}} \right|_{\mathbf{s}_{k}}.$$
(12)

[revised manuscript text omitted]